# SSG: Scaled Spatial Guidance for Multi-Scale Visual Autoregressive Generation

**Youngwoo Shin**[*]     **Jiwan Hur**[*]     **Junmo Kim**
Korea Advanced Institute of Science and Technology
{yshin0917, jiwan.hur, junmo.kim}@kaist.ac.kr

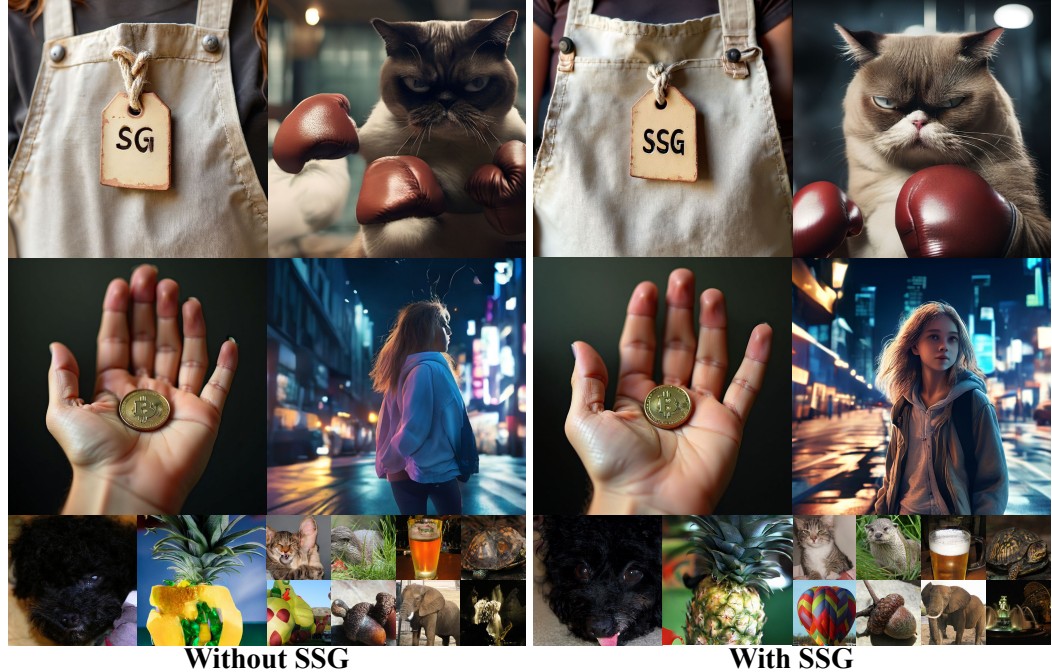

**Without SSG**                    **With SSG**

Figure 1: **SSG provides a training-free generation quality improvement for next-scale prediction models at negligible cost**, yielding sharper detail, fewer artifacts, and preserved global coherence. Full input prompts and model specifications are in Appx. G.

## Abstract

Visual autoregressive (VAR) models generate images through next-scale prediction, naturally achieving coarse-to-fine, fast, high-fidelity synthesis mirroring human perception. In practice, this hierarchy can drift at inference time, as limited capacity and accumulated error cause the model to deviate from its coarse-to-fine nature. We revisit this limitation from an information-theoretic perspective and deduce that ensuring each scale contributes high-frequency content not explained by earlier scales mitigates the train–inference discrepancy. With this insight, we propose Scaled Spatial Guidance (SSG), training-free, inference-time guidance that steers generation toward the intended hierarchy while maintaining global coherence. SSG emphasizes target high-frequency signals, defined as the semantic residual, isolated from a coarser prior. To obtain this prior, we leverage a principled frequency-domain procedure, Discrete Spatial Enhancement (DSE), which is devised to sharpen and better isolate the semantic residual through frequency-aware construction. SSG applies broadly across VAR models leveraging discrete visual tokens, regardless of tokenization design or conditioning modality. Experiments demonstrate SSG yields consistent gains in fidelity and diversity while preserving low latency, revealing untapped efficiency in coarse-to-fine image generation. Code is available at https://github.com/Youngwoo-git/SSG.

---

[*]Equal contribution.

# 1    INTRODUCTION

Visual Autoregressive (VAR) structured models generate images via a sequence of discrete visual tokens in a next-scale, coarse-to-fine paradigm, delivering highly competitive fidelity and diversity at substantial throughput (Tian et al., 2024; Tang et al., 2025; Han et al., 2025). Requiring only about a dozen inference steps, these models offer an efficient and conceptually grounded approach to visual synthesis that aligns with the hierarchical nature of human perception.

Improving VAR-structured models has been pursued along several axes: adding auxiliary refinement modules (Tang et al., 2025; Chen et al., 2025b; Kumbong et al., 2025), modifying the transformer architecture for generation (Voronov et al., 2025), modifying tokenization (Qu et al., 2025; Han et al., 2025), and replacing the native coarse-to-fine generation with flow matching (Ren et al., 2025; Liu et al., 2025). While these approaches push the boundary of VAR-structured models, they typically require costly retraining and introduce overhead, undermining the efficiency that motivates the VAR paradigm. Furthermore, they are susceptible to train-inference discrepancy caused by error accumulation. While several methods have been proposed to mitigate this issue (Chen et al., 2025b; Kumbong et al., 2025; Han et al., 2025), it remains a persistent challenge for VAR-structured models.

In this paper, we re-examine next-scale prediction in VAR from an information-theoretic perspective. Our analysis identifies a core principle that mitigates the train–inference discrepancy. Specifically, if each prediction step adds scale-appropriate novel information not captured by the previous scale, it reduces informational redundancy. This raises a central question: *How can we guide the model to add the intended novel information at each step, realigning VAR with its coarse-to-fine nature?*

To address this challenge, we propose **Scaled Spatial Guidance** (SSG), training-free guidance for VAR models with negligible overhead. SSG sets the target at each step to the *semantic residual*, the high-frequency detail specific to that scale. To isolate this residual from the coarse structure, we use a prior carrying that coarser structure from the preceding step. This prior is constructed via *Discrete Spatial Enhancement* (DSE), a frequency-domain interpolation that preserves structural integrity across scales. Together, these components promote principled progression from coarse structure to fine detail. SSG applies across VAR models with discrete visual tokens, independent of tokenization and conditioning, and significantly improves fidelity without additional data or fine-tuning.

We evaluate SSG on strong VAR baselines with varied tokenization (Tian et al., 2024; Tang et al., 2025; Han et al., 2025), achieving consistent gains on class- and text-conditional generation. Across different VAR scales, applying SSG yields robust and competitive performance relative to recent diffusion (Yan et al., 2024; Hatamizadeh et al., 2024; Peebles & Xie, 2023; Alpha-VLLM, 2024; Dhariwal & Nichol, 2021; Ho et al., 2022) and masked models (Chang et al., 2022; Li et al., 2024b), while preserving the low latency of VAR architectures.

Our contributions are as follows:

- We propose Scaled Spatial Guidance (SSG), training-free guidance that enforces a coarse-to-fine hierarchy by prioritizing the generation of novel, high-frequency information at each step.
- We reinterpret VAR sampling from an information-theoretic perspective and analyze the per-step objective, identifying the optimal priority at each step for robust generation.
- We demonstrate consistent improvements in both fidelity and diversity with negligible latency overhead, enhancing VAR-structured models for discrete visual generation.

# 2    METHODS

## 2.1    PRELIMINARIES: NEXT-SCALE AUTOREGRESSIVE GENERATION

The Visual Autoregressive (VAR) framework (Tian et al., 2024) re-frames autoregressive visual generation from conventional "next-token prediction" to a hierarchical, coarse-to-fine "next-scale prediction." This approach operates on an image represented as a sequence of hierarchical token maps, $(r_1, r_2, \ldots, r_K)$ (Esser et al., 2021; Lee et al., 2022; Tian et al., 2024), mirroring the human perceptual tendency to resolve global structures before fine-grained details.

Specifically, a feature map $f \in \mathbb{R}^{h \times w \times C}$ is quantized into $K$ discrete token maps, $(r_1, \ldots, r_K)$, of progressively finer resolutions. The generation of each map $r_k \in \{1, \ldots, V\}^{h_k \times w_k}$ is conditioned

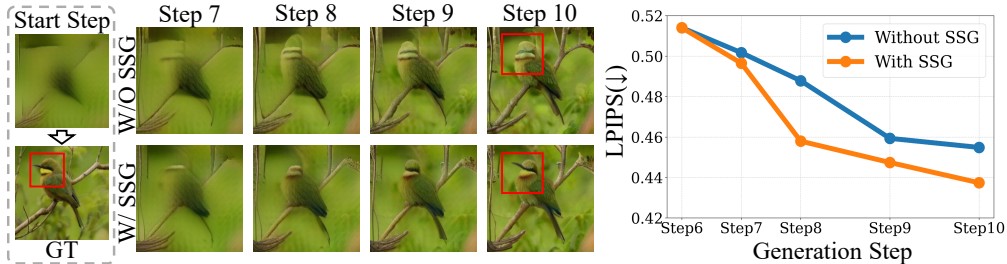

Figure 2: **Impact of SSG on Image Completion (VAR-d30). (Left)** By amplifying the semantic residual, SSG enables the model to accurately reconstruct high-frequency details like the bird's beak (red box), unlike the baseline. **(Right)** Consistently better LPIPS substantiates this improvement.

on the preceding maps $r_{<k} = (r_1, \ldots, r_{k-1})$, where $V$ is the codebook vocabulary size. The base map, $r_1$, contains the global context and is predicted from initial class or text tokens. The joint probability distribution is then factorized autoregressively across these scales:

$$p(r_1, r_2, \ldots, r_K) = p(r_1) \prod_{k=2}^{K} p(r_k \mid r_{<k}). \tag{1}$$

At each step $k$, a model $\mathcal{M}$ generates a residual logit tensor $\ell_k \in \mathbb{R}^{h_k \times w_k \times V}$ conditioned on $r_{<k}$, which defines a categorical distribution at each spatial location from which $r_k$ is sampled.

To synthesize an image, the generative process builds a final feature representation from the token maps $(r_1, \ldots, r_K)$ via residual de-quantization and accumulation (Lee et al., 2022; Tian et al., 2024). At each step $k$, the token map $r_k$ is de-quantized into a continuous residual feature map, $z_k$, using its corresponding codebook embedding. Each residual $z_k$ is then upsampled to the target resolution by an operator $U(\cdot)$ and added to an accumulated feature map: $\hat{f}_k = \hat{f}_{k-1} + U(z_k)$, with $\hat{f}_0 = \mathbf{0}$. Finally, the completed map $\hat{f}_K$ is passed to a decoder to produce the output image.

While powerful, the effectiveness of this multi-scale generative process requires the model to faithfully learn the hierarchical structure of the token representation, such as that from a multi-scale VQVAE (Tian et al., 2024). This representation is structured such that ideally each subsequent generative step $k$ exclusively models a new, higher-frequency band of details. In practice, however, limited model capacity prevents strict adherence to this hierarchical frequency separation. This deviation from the ideal behavior becomes a primary source of the train-inference discrepancy.

Consequently, the model often fails its designated role at each inference step. Instead of introducing novel, finer details, it redundantly predicts lower-frequency information already established in previous steps. This inefficient misallocation of model capacity, a direct result of the train-inference discrepancy, leads to the structural degradation and spatial disorientation seen in the upper row of Fig. 2. Therefore, the central challenge is to guide the generative process at each step $k$ to focus exclusively on synthesizing the novel, higher-frequency details appropriate for that step.

## 2.2 DERIVATION OF SCALED SPATIAL GUIDANCE

To analyze details added per step, we re-interpret VAR sampling as a variational optimization problem via the Information Bottleneck (IB) principle (Tishby et al., 2000; Alemi et al., 2017), to derive principled guidance to enhance fidelity by mitigating train-inference discrepancy. The IB principle seeks a compressed representation $\tilde{X}$ of an input $X$ maximally informative about a target $Y$:

$$\mathcal{L}_{\text{IB}} = \min_{\tilde{X}} I(X; \tilde{X}) - \beta I(\tilde{X}; Y), \tag{2}$$

where $I(\cdot; \cdot)$ denotes mutual information. For VAR's sequential generation, the IB principle is conceptually reversed: rather than compressing data, the goal at each step $k$ is to generate a residual $z_k$ adding new, finer details. Thus, the objective in Eq. (2) maximizes information about the final output $\hat{f}_K$ while minimizing redundancy with the previous state $\hat{f}_{k-1}$, yielding the VAR-specific objective:

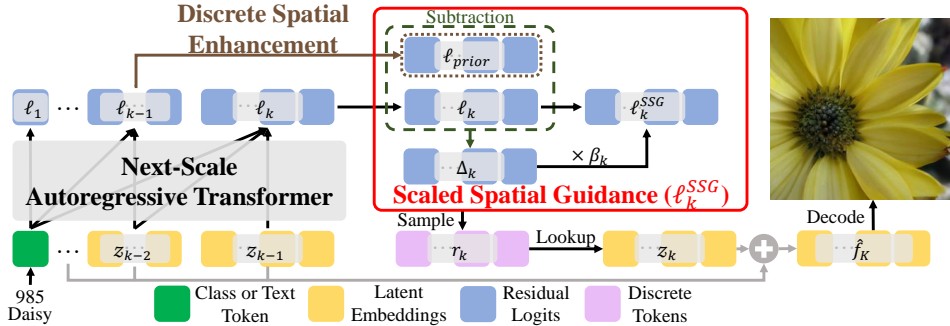

Figure 3: **Overview of a VAR-structured model** with our Scaled Spatial Guidance (SSG) module. At each step, the autoregressive transformer predicts residual logits, which SSG refines by using a DSE-enhanced prior to isolate and amplify the high-frequency semantic residual before sampling.

$$\mathcal{L}_{\text{VAR-IB}} = \max_{z_k} \beta I(z_k; \hat{f}_K | \hat{f}_{k-1}) - I(\hat{f}_{k-1}; z_k). \tag{3}$$

Expanding the conditional term via the chain rule of mutual information[1] yields:

$$\mathcal{L}_{\text{VAR-IB}} = \max_{z_k} \beta I(z_k; \hat{f}_K) - (\beta + 1) I(z_k; \hat{f}_{k-1}). \tag{4}$$

We further simplify this objective from a frequency-domain perspective. By decomposing the output $\hat{f}_K$ via ideal low-pass ($L$) and high-pass ($H$) filters into its low-frequency ($L(\hat{f}_K) \approx \hat{f}_{k-1}$) and high-frequency ($H(\hat{f}_K)$) components, the objective reduces to an intuitive form:

$$\mathcal{L}_{\text{VAR-IB}} \approx \max_{z_k} \beta I(z_k; H(\hat{f}_K)) - I(z_k; L(\hat{f}_K)). \tag{5}$$

To translate Eq. (5) into practice, we work at the logit level: the model samples a residual token $r_k$ from residual logits $\ell_k$, whose codebook embedding yields $z_k$. We therefore construct an IB-inspired, Maximum a Posteriori (MAP)-style surrogate with two complementary parts:

**Target-informativeness term** $\left( \beta I(z_k; H(\hat{f}_K)) \right)$ promotes adding new, fine-scale detail. Here $\ell_{\text{prior}}$ is a coarse reference carrying information from the previous step, and $\ell'$ is the guided version of the step-$k$ logits optimized for sampling. We encourage $\ell'$ to follow our proxy for high-frequency detail, the semantic residual $\Delta_k = \ell_k - \ell_{\text{prior}}$, via the dot-product surrogate $\beta (\ell')^\top \Delta_k$.

**State-redundancy term** $\left( - I(z_k; L(\hat{f}_K)) \right)$ limits deviation from established coarse structure. In practice, we use an L2 proximity regularizer that keeps the guided logits $\ell'$ close to the step-$k$ base logits $\ell_k$, adding the quadratic proximity term $-\frac{1}{2}\|\ell' - \ell_k\|_2^2$.

Combining these yields an objective conceptually aligned with the log-posterior of a MAP formulation (Appx. C). Optimizing this objective over the guided logits admits a closed-form solution:

$$\mathcal{L}(\ell') = \beta (\ell')^\top \Delta_k - \frac{1}{2}\|\ell' - \ell_k\|_2^2, \qquad \ell' \in \mathbb{R}^{|\mathcal{V}|}, \tag{6}$$

where $\ell_k \in \mathbb{R}^{|\mathcal{V}|}$ is the residual logits at step $k$, $\Delta_k \in \mathbb{R}^{|\mathcal{V}|}$ is the semantic residual, and $\beta \geq 0$. This quadratic is strictly concave in $\ell'$ (Hessian $-I$) with unique maximizer

$$\ell_k^{\text{SSG}} = \ell_k + \beta \Delta_k. \tag{7}$$

Letting $\beta$ be stepwise, $\beta_k$, yields **Scaled Spatial Guidance (SSG)** (full derivation in Appx. D).

$$\ell_k^{\text{SSG}} = \ell_k + \beta_k \Delta_k = \ell_k + \beta_k (\ell_k - \ell_{\text{prior}}). \tag{8}$$

---

[1]The coarse-state approximation and deterministic conditioning for the chain rule: see Appx. A, B.

The scaling factor $\beta_k$ controls the magnitude of the semantic residual $\Delta_k$, trading off the injection of high-frequency detail against preservation of base-model coherence. Empirically, SSG refines detailed structures (e.g., the bird's beak in Fig. 2) and yields consistently lower LPIPS across generation steps (graph in Fig. 2), in line with emphasizing the target-informativeness term while suppressing the state-redundancy term in Eq. (5). Nonetheless, the effect depends on the quality of the transported prior $\ell_{\text{prior}}$: if the prior is distorted, $\Delta_k$ can be misaligned and suppress essential detail. Thus, principled construction of $\ell_{\text{prior}}$ is critical to realizing the full benefit of SSG.

---

**Algorithm 1** DSE formulation

1: **Input:** Previous logits $\ell_{\text{prev}}$; target size $S_k$
2: **Output:** Upsampled prior $\ell_{\text{prior}}$
3: **if** $\ell_{\text{prev}}$ is not None **then**
4: $\quad \ell'_{\text{interp}} \leftarrow \text{Interpolate}(\ell_{\text{prev}}, S_k)$;
5: $\quad L_{\text{prev}} \leftarrow \text{DCT}(\ell_{\text{prev}})$;
6: $\quad L'_{\text{interp}} \leftarrow \text{DCT}(\ell'_{\text{interp}})$;
7: $\quad \tilde{L} \leftarrow L'_{\text{interp}}$;
8: $\quad \tilde{L}[0 : \text{size}(L_{\text{prev}})] \leftarrow L_{\text{prev}}$;
9: $\quad \ell_{\text{prior}} \leftarrow \text{IDCT}(\tilde{L})$;
10: $\quad$ **return** $\ell_{\text{prior}}$;
11: **end if**

---

**Algorithm 2** SSG Formulation

1: **Input:** Raw logits $\ell_k$; previous logits $\ell_{\text{prev}}$
2: **Hyperparameter:** Guidance scale $\beta_k$
3: **Initialize:** Guided logits $\ell_k^{\text{SSG}}$
4: **if** $k = 1$ **then**
5: $\quad \ell_k^{\text{SSG}} \leftarrow \ell_k$;
6: **else**
7: $\quad \ell_{\text{prior}} \leftarrow \text{DSE}(\ell_{\text{prev}}, \text{size}(\ell_k))$;
8: $\quad \Delta_k \leftarrow \ell_k - \ell_{\text{prior}}$;
9: $\quad \ell_k^{\text{SSG}} \leftarrow \ell_k + \beta_k \cdot \Delta_k$;
10: **end if**
11: **return** $\ell_k^{\text{SSG}}$

---

## 2.3 PRIOR CONSTRUCTION IN THE FREQUENCY DOMAIN

We construct the prior $\ell_{\text{prior}}$ from the previous step's logits $\ell_{k-1}$. Because the hierarchy is relative, $\ell_{k-1}$ encodes a coarser, lower-frequency band than the details at step $k$, but its smaller spatial scale requires upsampling. Simple spatial interpolation is local and approximate: linear interpolation yields an overly smooth, attenuated prior, while nearest neighbor introduces blocky discontinuities and spurious high frequencies, contaminating the semantic residual. In contrast, a frequency-domain construction leverages orthonormal discrete transforms to provide a global, energy-preserving representation in which bands are independent and non-interfering. This independence affords two benefits: precise separation in the forward transform and exact, lossless reconstruction in the inverse. As a result, coarse structure is preserved without distortion, enabling $\Delta_k$ to isolate the new information bandwidth required at step $k$.

To implement this, we introduce **Discrete Spatial Enhancement (DSE)**, performing spectral fusion in the frequency domain. DSE first applies a discrete frequency transform to two signals: the original coarse logits $\ell_{k-1}$ and a simple upscaled version, $\ell'_{\text{interp}}$. The low-frequency coefficients of the transformed $\ell_{k-1}$ serve as the ground-truth coarse structure, while the high-frequency coefficients of the transformed $\ell'_{\text{interp}}$ provide a plausible extrapolation of new details. We then construct a hybrid frequency spectrum by combining the low-frequency coefficients from the former with the high-frequency coefficients from the latter. Applying the inverse transform to this fused spectrum yields a prior, $\ell_{\text{prior}}$, that rigorously preserves the verbatim coarse structure from the original logits while incorporating a plausible high-frequency extrapolation. The full process is detailed in Alg. 1. In our implementation, the Discrete Cosine Transform (DCT) serves as the discrete frequency transform.

## 2.4 EFFICIENT INFERENCE-TIME IMPLEMENTATION

A key advantage of the SSG framework is its seamless integration into pretrained VAR-structured models at inference time. Our method operates directly on the residual logits, the pre-activation outputs defining discrete token probabilities. This makes it agnostic to the underlying model architecture, requiring no modifications to model weights or the introduction of new branches. Consequently, it is broadly applicable to any VAR-structured models that generate images with discrete tokens. Furthermore, its effectiveness is independent of the specific number of generative steps or the resolutions used, ensuring robust performance across diverse model configurations.

The computational overhead of this framework is negligible. As detailed in Alg. 2, DSE leverages the raw residual logits cached from the previous step, avoiding any extra forward passes. The entire SSG mechanism, consisting of the DSE step and a subsequent linear combination, can be implemented in just a few lines of code. With frequency domain operations adding only a minimal computational and memory cost, SSG enhances structural and semantic consistency while largely preserving the efficiency of the original pretrained model. This makes it a practical tool for improving both class-conditional and text-conditional generation without compromising on speed.

## 3  RELATED WORK

**Autoregressive models** build on VAEs (Kingma & Welling, 2013) by modeling discrete image tokens from tokenizers such as VQ-VAE (Van Den Oord et al., 2017) and VQGAN (Esser et al., 2021). Masked-prediction variants improve quality but incur significant inference compute cost (Li et al., 2024b). VAR (Tian et al., 2024) shifts from *next-token* to *next-scale* prediction, with progress in token design (hybrid (Tang et al., 2025), bit-wise (Han et al., 2025)), architecture (Li et al., 2025; Chen et al., 2025b; Voronov et al., 2025), and flow-matching integration (Ren et al., 2025; Liu et al., 2025). Despite these refinements, a core issue persists: a train–inference discrepancy whereby finite-capacity VAR generators fail to reliably realize the coarse-to-fine hierarchy implied by multi-scale tokenization at inference. Recent methods reduce this gap via refinement mechanisms, where CoDe (Chen et al., 2025b) adds a collaborative refiner, HMAR (Kumbong et al., 2025) performs multi-step masked prediction, and Infinity (Han et al., 2025) introduces bitwise self-correction with redefined tokenization, yet they require model modifications and retraining, increasing memory usage or latency. In contrast, **SSG** addresses the train–inference discrepancy at inference time: it promotes scale-specific novel detail while preserving established coarse structure, aligning VAR with its coarse-to-fine hierarchy without architectural changes, additional data, or significant overhead.

**Visual guidance** improves generation by sharpening the predictive distribution—akin to lowering temperature in language models, which reduces entropy and increases faithfulness at the cost of diversity (Tumanyan et al., 2023). However, existing techniques incur distinct trade-offs. Classifier-free guidance (CFG) can miss fine spatial details (Ho & Salimans, 2021); auto-guidance requires a second model (Karras et al., 2024); and autoregressive strategies like CCA require costly fine-tuning (Chen et al., 2025a). A separate family of diffusion controls, including SAG (Hong et al., 2023), PAG (Ahn et al., 2024), SDG (Feng et al., 2023), and STG (Hyung et al., 2025), provides granular conditioning but is not tailored to the coarse-to-fine structure of VAR frameworks and typically adds extra inference steps, increasing latency. In contrast, we introduce training-free guidance tailored to VAR-structured models that uses no additional data and adds negligible overhead.

## 4  EXPERIMENTS

We evaluate SSG via four questions: **(1)** Does it improve VAR models across scales to be competitive with other leading generative families? (Sec. 4.2) **(2)** Is it robust across advanced tokenization schemes? (Sec. 4.3) **(3)** Does it enhance high-frequency detail, as motivated in Sec. 2.2? (Sec. 4.4) **(4)** Is the frequency-domain DSE implementation effective, as discussed in Sec. 2.3? (Sec. 4.5)

Table 1: **Performance gains from SSG on VAR models across scales** on ImageNet 256×256.

| Model | FID↓ | sFID↓ | IS↑ | Pre↑ | Rec↑ | #Para | #Step | Time |
|---|---|---|---|---|---|---|---|---|
| VAR-d16 | 3.42 | 8.70 | 275.6 | 0.84 | **0.51** | 310M | 10 | 0.5 |
| +SSG (**Ours**) | **3.27** | **8.39** | **285.3** | **0.85** | 0.50 | 310M | 10 | 0.5 |
| VAR-d20 | 2.67 | 7.97 | 299.8 | **0.83** | 0.55 | 600M | 10 | 0.6 |
| +SSG (**Ours**) | **2.49** | **7.60** | **305.2** | **0.83** | **0.56** | 600M | 10 | 0.6 |
| VAR-d24 | 2.39 | 8.18 | 314.7 | 0.82 | 0.58 | 1.0B | 10 | 0.7 |
| +SSG (**Ours**) | **2.20** | **6.95** | **324.0** | **0.83** | **0.59** | 1.0B | 10 | 0.7 |
| VAR-d30 | 2.02 | 8.52 | 302.9 | **0.82** | 0.60 | 2.0B | 10 | 1.0 |
| +SSG (**Ours**) | **1.68** | **8.50** | **313.2** | 0.81 | **0.62** | 2.0B | 10 | 1.0 |

Table 2: **Visual Generative model comparison on ImageNet 256×256 benchmark.** Metrics include Fréchet inception distance (FID), inception score (IS), precision (Pre), and recall (Rec). Model parameters (#Para), inference steps (#Step), and inference time relative to VAR-d30 are reported. †: Taken from VAR (Tian et al., 2024). ‡: Taken from HART (Tang et al., 2025). §: Reproduced.

| Type | Model | FID↓ | IS↑ | Pre↑ | Rec↑ | #Para | #Step | Time |
|---|---|---|---|---|---|---|---|---|
| GAN | GigaGAN† (Kang et al., 2023) | 3.45 | 225.5 | 0.84 | 0.61 | 569M | 1 | – |
| | StyleGAN-XL† (Sauer et al., 2022) | 2.30 | 265.1 | 0.78 | 0.53 | 166M | 1 | 0.3 |
| Diff. | LDM-4-G† (Rombach et al., 2022) | 3.60 | 247.7 | – | – | 400M | 250 | – |
| | DiT-XL/2† (Peebles & Xie, 2023) | 2.27 | 278.2 | 0.83 | 0.57 | 675M | 250 | 45 |
| | L-DiT-7B† (Alpha-VLLM, 2024) | 2.28 | 316.2 | 0.83 | 0.58 | 7.0B | 250 | > 45 |
| | $D_{IFFU}$SSM-XL-G (Yan et al., 2024) | 2.28 | 259.1 | **0.86** | 0.56 | 660M | 250 | – |
| | DiffiT (Hatamizadeh et al., 2024) | 1.73 | 276.5 | 0.80 | **0.62** | 561M | 250 | – |
| Mask. | MaskGIT† (Chang et al., 2022) | 6.18 | 182.1 | 0.80 | 0.51 | 227M | 8 | 0.5 |
| | MAR-B‡ (Li et al., 2024b) | 2.31 | 281.7 | – | – | 208M | 64 | 10.0 |
| | MAR-H‡ (Li et al., 2024b) | 1.78 | 296.0 | – | – | 479M | 64 | 13.4 |
| AR | VQGAN† (Esser et al., 2021) | 18.65 | 80.4 | 0.78 | 0.26 | 227M | 256 | 19 |
| | RQTransformer† (Lee et al., 2022) | 7.55 | 134.0 | – | – | 3.8B | 68 | 21 |
| | GIVT-Causal-L+A (Tschannen et al., 2024) | 2.59 | – | 0.81 | 0.57 | 304M | 256 | – |
| | LlamaGen-3B (Sun et al., 2024) | 2.18 | 267.7 | 0.84 | 0.54 | 3.1B | 1 | – |
| VAR | VAR-CoDe N=9 (Chen et al., 2025b) | 1.94 | 296 | 0.80 | 0.61 | 2.3B | 10 | – |
| | HMAR-d30 (Kumbong et al., 2025) | 1.95 | **334.5** | 0.82 | **0.62** | 2.4B | 14 | – |
| | VAR-d30§ (Tian et al., 2024) | 2.02 | 302.9 | 0.82 | 0.60 | 2.0B | 10 | 1.0 |
| | +SSG (**Ours**) | **1.68** | 313.2 | 0.81 | **0.62** | 2.0B | 10 | 1.0 |

## 4.1 EXPERIMENTAL SETTINGS

We evaluate class-conditional ImageNet generation at $256 \times 256$ and $512 \times 512$ (Deng et al., 2009), primarily using Fréchet Inception Distance (FID) (Heusel et al., 2017) to assess fidelity and diversity, along with Inception Score (IS) (Salimans et al., 2016) and spatial FID (sFID) (Nash et al., 2021). For text-to-image (T2I), we use the MJHQ-30K benchmark (Li et al., 2024a) and assess semantic fidelity and prompt alignment with FID (Heusel et al., 2017) and CLIPScore (Hessel et al., 2021). We also report inference latency to quantify SSG's computational overhead across all models.

Our analysis focuses on VAR-structured models, which exemplify the next-scale paradigm (Tian et al., 2024; Tang et al., 2025; Han et al., 2025). To contextualize Tab. 2, we also compare against leading models from other paradigms: high-fidelity diffusion ($D_{IFFU}$SSM-XL-G (Yan et al., 2024), DiffiT (Hatamizadeh et al., 2024)), GANs (StyleGAN-XL (Sauer et al., 2022)), autoregressive (LlamaGen-3B (Sun et al., 2024)), and masked models (MAR-H (Li et al., 2024b)).

For a fair comparison, we report metrics with CFG enabled whenever supported. Reproducibility of VAR-family checkpoints posed challenges: for VAR (Tian et al., 2024), the released weights underperform the results reported in the paper; for HART (Tang et al., 2025), public discussions note difficulty matching reported scores; and Infinity lacks official MJHQ-30K results. Thus, we re-evaluated all VAR baselines on a single NVIDIA A6000 under a unified protocol, and all gains are measured by applying SSG to these runs under identical settings. The SSG strength follows a linear decay, $\beta_k = \beta\left(1 - \frac{k-1}{K}\right)$ (Sec. 2.2), where $\beta$ is the initial scale and $K$ is the number of steps.

## 4.2 TRAINING-FREE ENHANCEMENT OF NEXT-SCALE GENERATIVE MODELS

Evaluating SSG across scaled VAR models reveals consistent performance gains that amplify with model capacity (Tab. 1). On class-conditional ImageNet $256 \times 256$, the FID reduction grows from 0.15 for VAR-d16 to a substantial 0.34 for the larger VAR-d30. Crucially, these improvements are achieved without altering model parameters or increasing inference latency. This confirms SSG achieves a scalable enhancement, improving with the representational power of the base model. This scaling trend culminates in our result on VAR-d30 (Tab. 2), where SSG achieves an FID of 1.68, surpassing competitors including DiffiT (1.73 at 256 steps) and MAR-H (1.78 at 64 steps; 13.4× slower).

While methods like HMAR-d30 achieve a higher IS through costly retraining and architectural modifications, SSG improves the baseline IS without modification to the pretrained model. This demonstrates the primary strength of SSG: achieving superior fidelity with significant efficiency by enhancing, not replacing, the original model.

The effectiveness of SSG extends to $512\times512$ resolution (Tab. 3), where it improves VAR-d36, reducing FID by 11.5% to 2.39 while increasing IS by 10.3% to a class-leading 320.6. While MAR-L attains a lower FID (1.73), it does so at a prohibitive cost, with an estimated inference time $\sim214\times$ longer than our SSG-enhanced VAR. This performance surpasses VAR variants like HMAR-d24 and diffusion baselines such as DiffiT. By mitigating the train–inference discrepancy and improving spatial coherence (Sec. 2.1), SSG offers a superior performance-efficiency trade-off.

Table 3: **ImageNet** $512 \times 512$ **conditional generation.** Inference times are relative to VAR-d36. †: Quoted from VAR. ‡: Estimated via linear scaling of steps ($4\times$) and pixels ($4\times$) from the reported time of the $256\times256$ model. §: Reproduced.

| Type | Model | FID↓ | IS↑ | Time |
|------|-------|------|-----|------|
| GAN | BigGAN† | 8.43 | 177.9 | – |
| Diff. | DiT-XL/2† | 3.04 | 240.8 | 81 |
| | $D_{IFFU}$SSM-XL-G | 3.41 | 255.1 | – |
| | DiffiT | 2.67 | 252.1 | – |
| Mask. | MaskGIT† | 7.32 | 156.0 | 0.5 |
| | MAR-L | **1.73** | 279.9 | 214.4‡ |
| AR | VQGAN† | 26.52 | 66.8 | 25 |
| VAR | HMAR-d24 | 2.99 | 304.1 | – |
| | VAR-d36§ | 2.70 | 290.6 | 1.0 |
| | +SSG (**Ours**) | 2.39 | **320.6** | 1.0 |

### 4.3 GENERALIZATION ACROSS DIVERSE TOKEN ARCHITECTURES

To demonstrate the generalizability of SSG, we first test it on a text-conditioned model with a different token structure: HART-0.7B, which uses hybrid continuous-discrete tokens. As shown in Tab. 4, SSG improves FID by 13.9% while maintaining a stable CLIPScore. This confirms that our method enhances spatial fidelity while preserving semantic alignment.

We further challenge SSG on Infinity-2B, a model with both bit-wise tokenization and a built-in bit-wise self-correction (BSC) mecha-

Table 4: **T2I Comparison using MJHQ30K**

| Model | FID↓ | CLIP Score↑ | Time (s) |
|-------|------|-------------|----------|
| HART-0.7B | 8.46 | 0.2819 | 1.06 |
| +SSG (**Ours**) | **7.28** | **0.2834** | 1.07 |
| Infinity-2B | 10.01 | 0.2754 | 1.83 |
| +SSG (**Ours**) | **9.68** | **0.2767** | 1.86 |

nism to mitigate teacher-forcing. SSG still delivers a 3.3% FID improvement with a stable CLIPScore. This result confirms the benefits of SSG are orthogonal to such model-specific corrections and validates its role in addressing the core train-inference discrepancy of VAR-structured models.

The versatility demonstrated on both hybrid and bit-wise tokens stems from the core design of SSG: it operates on the universal, pre-quantization logit space, making it agnostic to the token structure. By enhancing spatial fidelity while preserving semantic integrity across diverse architectures, SSG is a fundamental and broadly applicable enhancement to the coarse-to-fine generation paradigm.

### 4.4 ANALYZING THE SCALE-WISE REFINEMENT MECHANISM

We empirically assess the role of SSG as a scale-wise refinement mechanism by analyzing the spectra of residual logits from VAR-d16. Figure 4(a) plots the relative change in the log-magnitude of Fourier-transformed latents under SSG, revealing a threshold at the Nyquist frequency of the previous step. Above it, SSG increases spectral energy to synthesize novel high-frequency details; below it, SSG suppresses redundant low-frequency updates as the curve stays near or below zero (green line). This redistribution empirically supports the refinement mechanism in Sec. 2.2.

We evaluate the impact of SSG on the quality–diversity trade-off by sweeping sampling temperatures for VAR-d16 and plotting FID vs. IS (Fig. 4b). Across the sweep, SSG demonstrates strong robustness by consistently improving the Pareto frontier: at comparable IS it attains lower FID, and at comparable FID it attains higher IS, achieving the lowest FID and the highest IS observed. This indicates that SSG improves peak fidelity and maximum diversity without degrading the trade-off.

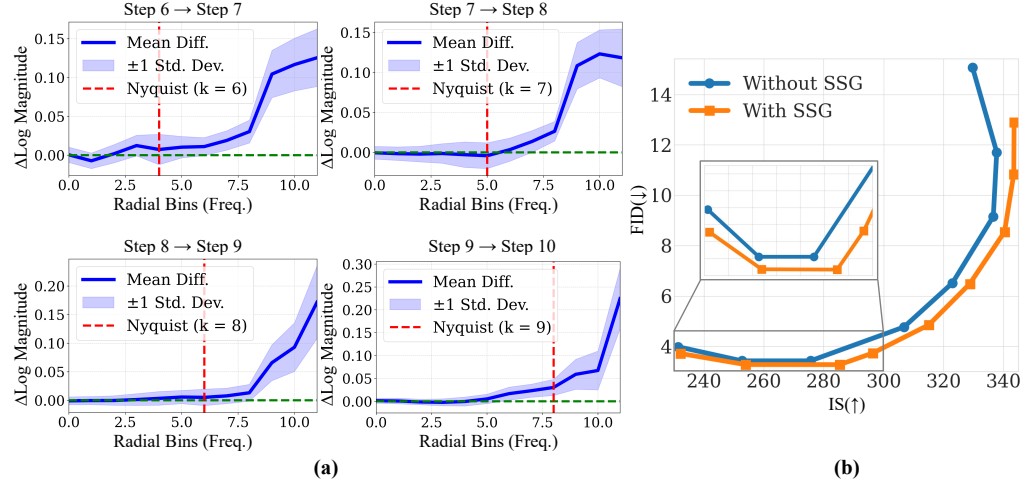

Figure 4: **Frequency-Domain Refinement and Performance. (a)** Analysis of the $\Delta$ log magnitude of Fourier-transformed latent embeddings. SSG redistributes the spectral energy by suppressing redundant low frequencies while selectively boosting the essential high-frequency energy beyond the Nyquist frequency (red line). **(b)** SSG achieves a consistently better FID vs. IS trade-off across sampling temperatures, indicating an improved quality-diversity profile. See Fig. 8 for the full trade-off graph over all evaluated sampling temperatures.

Table 5: **Ablation of SSG on VAR-d16, covering expansion type and $\ell_{\mathbf{prior}}$ formulation.** †: baseline without SSG implementation; ‡: zero padding replaces extrapolation from $L'_{\text{interp}}$.

| Expansion Type | $\ell_{\mathbf{prior}}$ Formulation | $\beta_k$ **Decay Schedule** | FID↓ | IS↑ | Relative Latency |
|---|---|---|---|---|---|
| Baseline† | N/A | N/A | 3.42 | 275.6 | 1.0 |
| Spatial | Nearest Neighbor | ✓ | 4.02 | 229.1 | 1.0 |
| Spatial | Linear | ✓ | 3.79 | 234.8 | 1.0 |
| Frequency | DSE‡ | ✓ | 3.34 | 277.6 | 1.0 |
| Frequency | DSE | ✗ | 3.63 | **287.8** | 1.0 |
| **Frequency** | **DSE (Ours)** | ✓ | **3.27** | 285.3 | 1.0 |

### 4.5 ABLATION STUDIES

**Prior formulation.** Tab. 5 contrasts the baseline (no SSG) with spatial- and frequency-domain formulations of $\ell_{\text{prior}}$ on VAR-d16. Spatial priors (nearest, linear) underperform the baseline in both FID and IS. Switching to frequency-domain DSE improves results: DSE$^{\dagger}$ achieves FID 3.34 and IS 277.6, surpassing both baseline and spatial variants. Our full DSE prior yields the best balance (FID 3.27, IS 285.3) at unchanged latency, supporting the frequency-domain design.

**Decay schedule.** A fixed $\beta_k$ (no decay) causes overguidance, producing exaggerated features recognizable to Inception yet off-distribution. This raises IS to 287.8 while degrading FID to 3.63. A linear decay schedule, however, stabilizes refinement and achieves a superior trade-off, yielding our best FID of 3.27 while maintaining a high IS of 285.3. See Appx. F for further $\beta_k$ scaling analysis.

## 5 CONCLUSION

We present **Scaled Spatial Guidance (SSG)**, training-free, logit-space guidance for VAR-structured models. SSG amplifies a semantic residual formulated with a frequency-domain prior using DSE, mitigating the train–inference discrepancy and reinforcing the intended coarse-to-fine hierarchy. Across VAR baselines and tokenizers, SSG delivers consistent gains in fidelity and diversity with negligible latency, competitive with or surpassing recent diffusion and masked models. We expect SSG to be a simple, model-agnostic building block for future work in next-scale generation.

## ETHICS STATEMENT

Scaled Spatial Guidance (SSG) is an inference-time technique that enhances pretrained generative models. While it can improve fidelity and controllability, the same capabilities could be misused by unauthorized actors. Risks include making deceptive or misleading media more convincing, with potential harms to privacy, reputation, and public trust.

Because SSG operates on existing models, it inherits their capabilities and limitations, including biases and harmful content patterns present in the underlying data. Our experiments therefore rely on publicly available, well-established models that include safety filters and community-vetted usage policies. SSG is not a safety filter itself; it should be deployed only alongside robust prompt and output moderation, provenance signals where appropriate, and human oversight for sensitive uses.

This work is intended for academic research and constructive applications. We explicitly prohibit malicious or unethical use, including the generation of deceptive content or content intended to cause harm. We encourage careful documentation of assumptions, adherence to model licenses and safety settings, and the development of clear ethical guidelines to ensure the responsible advancement of guidance methods and the broader generative modeling community.

## REPRODUCIBILITY STATEMENT

We are committed to ensuring the reproducibility of our research. To facilitate this, we will make our source code for Scaled Spatial Guidance (SSG) publicly available. The appendix provides comprehensive implementation details, including prompts used across experiments, hyperparameter settings for all experiments, and the specific publicly available pretrained models used in our evaluation.

## ACKNOWLEDGMENTS

This work was supported by Institute of Information & communications Technology Planning & Evaluation (IITP) grant funded by the Korea government(MSIT) (No.RS-2022-II220184, Development and Study of AI Technologies to Inexpensively Conform to Evolving Policy on Ethics).

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

APPENDIX

## A COARSE-STATE APPROXIMATION AND FREQUENCY HEURISTIC

We assume the established coarse structure satisfies $\hat{f}_{k-1} \approx L(\hat{f}_K)$ and that $I(z_k; \hat{f}_{k-1} \mid L(\hat{f}_K)) \leq \varepsilon$ for small $\varepsilon$ (approximate stepwise sufficiency). The low/high-frequency split leveraging ideal low pass filter (L) and high pass filter (H) ($L + H = \text{Id}$) is used for intuition; by Data Processing Inequality (DPI), filtering can only reduce Mutual Information (MI).

## B EXPANSION OF THE VAR-IB OBJECTIVE

Here, we provide a detailed derivation for the expansion of the VAR-specific Information Bottleneck objective. We begin with the objective as defined in the main text:

$$\mathcal{L}_{\text{VAR-IB}} = \max_{z_k} \beta I(z_k; \hat{f}_K \mid \hat{f}_{k-1}) - I(\hat{f}_{k-1}; z_k) \tag{9}$$

The simplification uses the expansion of the conditional mutual information term, $I(z_k; \hat{f}_K \mid \hat{f}_{k-1})$. We leverage the chain rule for mutual information (Cover & Thomas, 2006), which is expressed as:

$$I(A; B \mid C) = I(A; B, C) - I(A; C) \tag{10}$$

This expression is applicable when the variables form a Markov chain $A \rightarrow B \rightarrow C$. This condition implies that $C$ is conditionally independent of $A$ given $B$, which simplifies the joint mutual information term $I(A; B, C)$ to $I(A; B)$. In our context, the variables are $A = z_k$, $B = \hat{f}_K$, and $C = \hat{f}_{k-1}$. The required Markov chain is therefore $z_k \rightarrow \hat{f}_K \rightarrow \hat{f}_{k-1}$. This Markov condition holds if our coarse state term $\hat{f}_{k-1}$ is a deterministic function of the final, high-resolution output $\hat{f}_K$ (i.e., $\hat{f}_{k-1} = L(\hat{f}_K)$). This leads us to elaborate on deterministic conditioning.

**Deterministic conditioning (exact chain rule).** Let $C_k := L(\hat{f}_K)$ denote the low-pass projection of the final output. Since $C_k$ is a deterministic function of $\hat{f}_K$, the chain rule holds *exactly*:

$$I(z_k; \hat{f}_K \mid C_k) = I(z_k; \hat{f}_K) - I(z_k; C_k). \tag{11}$$

Substituting $C_k$ as the established coarse structure yields

$$\mathcal{L}_{\text{VAR-IB}} = \max_{z_k} \beta I(z_k; \hat{f}_K) - (\beta + 1) I(z_k; C_k).$$

To connect with the VAR state, we use the coarse-state approximation $\hat{f}_{k-1} \approx C_k$ (see Appx. A). With the low/high-frequency decomposition $\hat{f}_K = L(\hat{f}_K) + H(\hat{f}_K)$, where $L(\cdot)$ and $H(\cdot)$ are deterministic filters (hence $I(\cdot; L(\hat{f}_K))$ and $I(\cdot; H(\hat{f}_K))$ are well-defined and non-increasing by DPI), identifying $C_k = L(\hat{f}_K)$ yields the intuitive form used in the main text.

Therefore, using the exact identity in Eq. (11) and the coarse-state approximation $\hat{f}_{k-1} \approx L(\hat{f}_K)$ (Appx. A), the conditional term satisfies

$$I(z_k; \hat{f}_K \mid \hat{f}_{k-1}) \approx I(z_k; \hat{f}_K) - I(z_k; \hat{f}_{k-1}). \tag{12}$$

Substituting this back into the objective and collecting terms yields

$$\mathcal{L}_{\text{VAR-IB}} \approx \max_{z_k} \beta \left( I(z_k; \hat{f}_K) - I(z_k; \hat{f}_{k-1}) \right) - I(\hat{f}_{k-1}; z_k)$$
$$= \max_{z_k} \beta I(z_k; \hat{f}_K) - \beta I(z_k; \hat{f}_{k-1}) - I(z_k; \hat{f}_{k-1})$$
$$= \max_{z_k} \beta I(z_k; \hat{f}_K) - (\beta + 1) I(z_k; \hat{f}_{k-1}).$$

## C  MAP INTERPRETATION OF THE SURROGATE

### C.1  STOCHASTIC-CHANNEL JUSTIFICATION OF THE DOT-PRODUCT SURROGATE

**Where randomness enters.**  At step $k$ we sample $r_k \sim \mathrm{Cat}(q')$ with $q' = \mathrm{softmax}(\ell'/T)$; then $z_k = \mathrm{emb}(r_k)$ and $\hat{f}_k = g(\hat{f}_{k-1}, z_k)$ are deterministic. Hence shaping $\ell'$ shapes the stochastic node.

**First-order IB-aligned ascent.**  Consider the power-tilted contrast

$$\mathcal{C}_s(q') \;=\; (1+s)\,\mathbb{E}_{q'}[\log p_\theta(r \mid c_k)] \;-\; s\,\mathbb{E}_{q'}[\log p_{\mathrm{prior}}(r \mid \hat{f}_{k-1})],$$

with logits $\ell_k$ and $\ell_{\mathrm{prior}}$ for the two heads. Evaluated at $q' = \mathrm{softmax}(\ell_k/T)$,

$$\nabla_{\ell'}\mathcal{C}_s\big(\mathrm{softmax}(\ell'/T)\big)\Big|_{\ell'=\ell_k} \;=\; \tfrac{s}{T}\,\Delta_k \quad \text{(up to a mean shift removable by softmax invariance)},$$

so a small logit update $\delta$ obeys $\mathcal{C}_s \approx \mathrm{const} + \tfrac{s}{T}\,\delta^\top \Delta_k$. Adding a quadratic proximity term $-\tfrac{1}{2}\|\delta\|_2^2$ yields

$$\max_\delta \; \tfrac{s}{T}\,\delta^\top\Delta_k - \tfrac{1}{2}\|\delta\|_2^2 \quad \Rightarrow \quad \delta^\star = \tfrac{s}{T}\,\Delta_k, \;\; \ell' = \ell_k + \beta\,\Delta_k \;(\beta = s/T),$$

which is the SSG update. Thus the dot product $(\ell')^\top \Delta_k$ is the natural first-order ascent direction for the categorical sampling channel.

**Robustness to logit preprocessing (DSE, temperature).**  In practice, the base/prior logits may be obtained after deterministic preprocessing: $\tilde{\ell}_k = P_k(\ell_k)$, $\tilde{\ell}_{\mathrm{prior}} = P_{k-1}(\ell_{\mathrm{prior}})$, e.g., frequency-aware interpolation (DSE) for the prior or temperature rescaling. Since sampling remains $r_k \sim \mathrm{Cat}(\mathrm{softmax}(\tilde{\ell}'/T))$, stochasticity still enters only via the categorical, and the first-order derivation applies with the processed novelty direction $\tilde{\Delta}_k = \tilde{\ell}_k - \tilde{\ell}_{\mathrm{prior}}$. Scalar rescalings (temperature) reparameterize $\beta$ via $\beta = s/T$.

More generally, for a locally linear map $\tilde{\ell}' \approx J\,\ell'$ around $\ell_k$, the quadratic proximal step becomes

$$\max_\delta \; \tfrac{s}{T}\,\delta^\top J^\top \tilde{\Delta}_k \;-\; \tfrac{1}{2}\,\delta^\top M\,\delta, \quad M \succeq 0,$$

with solution $\delta^\star = \tfrac{s}{T}\,M^{-1}J^\top\tilde{\Delta}_k$. Choosing $M = I$ (our L2 proximity) and $J \approx I$ recovers $\ell' = \ell_k + \beta\,\tilde{\Delta}_k$. Thus DSE-based construction of $\ell_{\mathrm{prior}}$ and temperature modify the effective direction and step size but do not alter the stochastic-channel justification or the closed-form SSG update.

### C.2  PROXIMITY REGULARIZATION: L2 VS. DISTRIBUTIONAL TRUST REGIONS

Our state-redundancy term uses an L2 proximity regularizer on logits, $-\tfrac{1}{2}\|\ell' - \ell_k\|_2^2$. Two remarks:

**Tikhonov view.**  This is a Tikhonov (weight-decay–style) trust region in logit space that stabilizes updates and yields the closed-form solution $\ell' = \ell_k + \beta\,\Delta_k$.

**Distributional alternative.**  One can instead impose a distributional trust region via a KL penalty between the base distribution $q_k = \mathrm{softmax}(\ell_k/T)$ and the guided distribution $q' = \mathrm{softmax}(\ell'/T)$, e.g.,

$$-\lambda\,\mathrm{KL}\big(q_k \parallel q'\big) \quad \text{or} \quad -\lambda\,\mathrm{KL}\big(q' \parallel q_k\big).$$

This aligns the constraint in probability space but generally eliminates the simple closed form for $\ell'$ and requires iterative updates. For small steps, a second-order expansion of KL around $\ell_k$ reduces to a quadratic in $\ell' - \ell_k$, recovering an L2-type proximal form (up to a positive semidefinite metric induced by the softmax Fisher information). We adopt the L2 surrogate for its simplicity and closed-form optimizer while noting KL-based trust regions as a compatible alternative.

### C.3 MAP INTERPRETATION

We then can view the guided logits $\ell'$ as obtained by MAP:

$$\log p(\ell' \mid \text{evidence}) \;=\; \underbrace{\beta\,(\ell')^{\top}\Delta_k}_{\text{log-likelihood surrogate}} \;+\; \underbrace{\log p(\ell')}_{\text{log prior}}, \quad p(\ell') \propto \exp\!\big(-\tfrac{1}{2}\|\ell' - \ell_k\|_2^2\big).$$

The likelihood surrogate $\propto \exp(\beta\,(\ell')^{\top}\Delta_k)$ rewards alignment with the novelty direction $\Delta_k = \ell_k - \ell_{\text{prior}}$, while the Gaussian prior anchors $\ell'$ near the base logits $\ell_k$. Maximizing the log-posterior gives exactly

$$\mathcal{L}(\ell') \;=\; \beta\,(\ell')^{\top}\Delta_k \;-\; \tfrac{1}{2}\|\ell' - \ell_k\|_2^2,$$

## D FULL DERIVATION OF SCALED SPATIAL GUIDANCE

We begin with the Information Bottleneck (IB) objective, which seeks a compressed representation $\tilde{X}$ of an input $X$ that is maximally informative about a target $Y$:

$$\mathcal{L}_{\text{IB}} \;=\; \min_{\tilde{X}} \; I(X;\tilde{X}) \;-\; \beta\,I(\tilde{X};Y), \tag{13}$$

where $I(\cdot\,;\cdot)$ denotes mutual information and $\beta > 0$ trades off compression and relevance.

**Instantiation for VAR at step $k$.** For sequential coarse-to-fine generation, set $X = \hat{f}_{k-1}$ (previous state), $\tilde{X} = z_k$ (residual to be generated), and $Y = \hat{f}_K$ (final output). Since we care about *novel* information about $\hat{f}_K$ beyond $\hat{f}_{k-1}$, we use conditional mutual information, yielding

$$\mathcal{L}_{\text{VAR-IB}} \;=\; \max_{z_k} \; \beta\,I\big(z_k;\hat{f}_K \mid \hat{f}_{k-1}\big) \;-\; I\big(\hat{f}_{k-1};z_k\big). \tag{14}$$

**Chain-rule simplification.** Under deterministic conditioning of the coarse state (Appx. A, B), $I(A;B \mid C) = I(A;B) - I(A;C)$ with $C$ a deterministic function of $B$. Since $\hat{f}_{k-1}$ is an approximately deterministic low-pass of $\hat{f}_K$,

$$\mathcal{L}_{\text{VAR-IB}} = \max_{z_k} \; \beta\big[I(z_k;\hat{f}_K) - I(z_k;\hat{f}_{k-1})\big] - I(z_k;\hat{f}_{k-1})$$

$$= \max_{z_k} \; \beta\,I(z_k;\hat{f}_K) \;-\; (\beta+1)\,I(z_k;\hat{f}_{k-1}). \tag{15}$$

**Frequency-domain reduction.** Decompose the final output into ideal low- and high-frequency components, $\hat{f}_K = L(\hat{f}_K) + H(\hat{f}_K)$. Approximating additivity of information across disjoint bands, $I(z_k;\hat{f}_K) \approx I(z_k;L(\hat{f}_K)) + I(z_k;H(\hat{f}_K))$, and identifying the coarse state with the low-frequency part, $\hat{f}_{k-1} \approx L(\hat{f}_K)$, we obtain the full intermediate steps:

$$\mathcal{L}_{\text{VAR-IB}} \approx \max_{z_k} \; \beta\Big(I\big(z_k;L(\hat{f}_K)\big) + I\big(z_k;H(\hat{f}_K)\big)\Big) \;-\; (\beta+1)\,I\big(z_k;L(\hat{f}_K)\big) \tag{16}$$

$$= \max_{z_k} \; \beta\,I\big(z_k;L(\hat{f}_K)\big) + \beta\,I\big(z_k;H(\hat{f}_K)\big) \;-\; \beta\,I\big(z_k;L(\hat{f}_K)\big) \;-\; I\big(z_k;L(\hat{f}_K)\big) \tag{17}$$

$$= \max_{z_k} \; \beta\,I\big(z_k;H(\hat{f}_K)\big) \;+\; \big(\beta - \beta - 1\big)\,I\big(z_k;L(\hat{f}_K)\big) \tag{18}$$

$$= \max_{z_k} \; \beta\,I\big(z_k;H(\hat{f}_K)\big) \;-\; I\big(z_k;L(\hat{f}_K)\big). \tag{19}$$

Thus, the ideal residual $z_k$ should be informative about new high-frequency content while uninformative about already-established low-frequency structure.

**Logit-level surrogate and closed-form guidance.** At step $k$, the model samples a residual token $r_k$ from residual logits $\ell_k \in \mathbb{R}^{|\mathcal{V}|}$; its embedding yields $z_k$. We construct a MAP-style surrogate aligned with Eq. (19) with two parts: (i) a target-informativeness term that follows a proxy for high-frequency detail, the *semantic residual* $\Delta_k := \ell_k - \ell_{\text{prior}}$, where $\ell_{\text{prior}}$ carries coarse information

from the previous step; and (ii) a state-redundancy penalty that keeps guided logits close to the base $\ell_k$. For guided logits $\ell'$,

$$\mathcal{L}(\ell') \;=\; \beta\,(\ell')^\top \Delta_k \;-\; \tfrac{1}{2}\|\ell' - \ell_k\|_2^2, \qquad \ell' \in \mathbb{R}^{|\mathcal{V}|}. \tag{20}$$

The objective is strictly concave in $\ell'$ (Hessian $-I$) and admits a unique maximizer obtained by setting the gradient to zero:

$$\nabla_{\ell'}\mathcal{L}(\ell') = \beta\,\Delta_k - (\ell' - \ell_k) \;=\; 0 \;\implies\; \ell' \;=\; \ell_k + \beta\,\Delta_k. \tag{21}$$

**Scaled Spatial Guidance.** Allowing the trade-off to vary by step, $\beta \mapsto \beta_k$, yields the SSG update

$$\ell_k^{\mathrm{SSG}} \;=\; \ell_k + \beta_k\,\Delta_k \;=\; \ell_k + \beta_k\,(\ell_k - \ell_{\mathrm{prior}}). \tag{22}$$

This closed-form guidance mirrors the high- vs. low-frequency information trade-off in Eq.( 19)while incurring negligible overhead.

Table 6: **Infinity Table,** latency measured for generating with batch size=1

| Method | FID↓ | ImageReward↑ | CLIP Score↑ | HPSv2.1↑ | GenEval↑ | Latency(s) |
|---|---|---|---|---|---|---|
| Infinity-2B | 10.01 | 0.952 | 0.275 | 30.46 | 0.683 | 1.83 |
| +SSG **(Ours)** | **9.68** | **0.964** | **0.277** | **30.61** | **0.690** | 1.86 |

# E  ADDITIONAL MODEL EVALUATION

In this section, we additionally report metrics that reflect human preference and prompt alignment: ImageReward (Xu et al., 2023), a reward model trained on human preferences; HPSv2.1 (Wu et al., 2023), a score for aesthetic quality and prompt alignment; and Geneval (Ghosh et al., 2023), a multi-dimensional benchmark for generative model evaluation. Also, we re-report FID and CLIP Score from Tab. 4. Overall, adding SSG to the baseline Infinity model provides overall improvement in all metrics, while adding only a minimal latency overhead. The detailed result is in Tab. 6.

# F  ANALYSIS OF GUIDANCE PARAMETER SCALING

This section analyzes the trade-off between key generation metrics. We vary the guidance parameter $\beta_k$ and plot the FID vs. IS to examine the balance between distribution fidelity and sample quality.

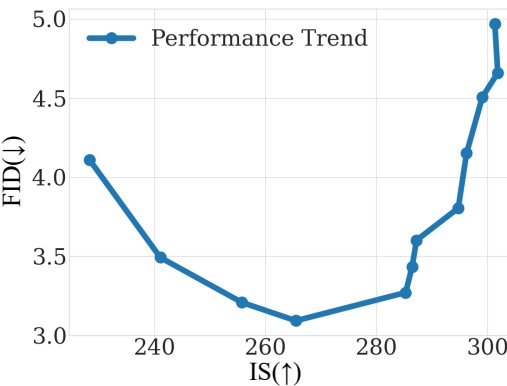

**Metric Trade-offs.** The plot on the left reveals a clear trade-off between FID and IS. Initially, increasing the guidance strength improves both metrics, achieving an optimal point. However, further pushing for higher IS values beyond this point leads to a sharp degradation in FID, indicating a loss in overall sample diversity and fidelity. We test $\beta_k$ values over the range [0.2, 2.4] with a step size of 0.2.

Figure 5: **The trade-off between FID and IS of the guidance parameter** $\beta_k$**.** The curve illustrates that optimizing solely for IS can be detrimental to the generation quality as measured by FID.

The results in Fig. 5 were obtained by applying SSG to the VAR-d16 model. To ensure an optimal balance, we select the $\beta_k$ from the point just before the FID score begins to degrade significantly.

| Prompt/Class Conditions | | | | | Models | | | | | |
|---|---|---|---|---|---|---|---|---|---|---|
| Artisan studio style. 'SSG' logo stamped into a small, imperfect ceramic tag, tied to the front of a clay-smudged canvas apron. Organic, wabi-sabi, handcrafted. | | | Grumpy cat as a boxer. | | Infinity-2B-1024px | | HART-7B-1024px | | | |
| A human palm with a coin | | | A photo of a teen girl walking on a city street at night, street photography, 4K, ultra HD, 3D shading beautiful, radiant, unity 3d, detailed, realistic, 3d shading, natural lighting. | | Infinity-2B-1024px | | HART-7B-1024px | | | |
| 265, Toy poodle | 953, Pineapple | 281, Tabby Cat | 360, Otter | 441, Beer Glass | 37, Box Turtle | VAR-d36 | VAR-d36 | VAR-d30 | VAR-d24 | VAR-d20 | VAR-d16 |
| | | 417, Balloon | 988, Acorn | 386, African Elephant | 562, Fountain | | | VAR-d30 | VAR-d24 | VAR-d20 | VAR-d16 |

Figure 6: Prompt and class used to generate Fig. 1, and exact model used leveraging SSG per image.

## G  DETAILED PROMPTS AND SPECIFICATIONS FOR FIG. 1

This appendix provides the exact prompts and class conditions used to generate the images in Fig. 1. We report both class-conditional and text-conditional models, evaluated at resolutions from 256×256 to 1024×1024. Model specifications are summarized in Fig. 6 for reproducibility. Display size in Fig. 1 is proportional to native resolution; a 256×256 image occupies one quarter of the area of a 1024×1024 image.

## H  SPECTRAL FIDELITY AND HIGH-FREQUENCY ROBUSTNESS

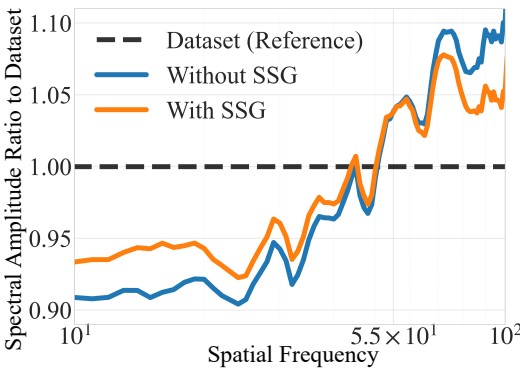

Figure 7: **Spectral Amplitude Ratio Analysis.** Images generated with SSG consistently adhere better to the distribution of the reference dataset.

To rigorously verify the perceptual impact of SSG, we extend our analysis to the pixel level. We compute average spectral energy profiles using 50,000 samples generated by VAR-d16 with and without SSG, comparing them against the 10,000 ImageNet validation images used for metrics in Tab. 1 to ensure statistical robustness. The resulting spectral analysis in Fig. 7 focuses on the frequency range $10^1$ to $10^2$, corresponding to meaningful fine textures rather than basic structure or extreme noise. In the band below $5.5 \times 10^1$, SSG consistently exhibits higher spectral energy than the baseline, effectively enhancing fine details. Crucially, at frequencies beyond $5.5 \times 10^1$, the baseline diverges from the reference curve, suggesting the possible amplification of artifacts or noise. In contrast, SSG maintains tighter alignment with the reference dataset, demonstrating that it regulates the generation process to match the true distribution rather than blindly amplifying noise.

## I  EXTENSION TO OTHER HIERARCHICAL GENERATIVE MODELS

While SSG is intentionally tailored for the explicit multiscale hierarchy of VAR, the underlying information-theoretic perspective introduced in Sec. 2.2 is not inherently restricted to this architecture; it holds potential for broader coarse-to-fine generative frameworks, such as diffusion and other autoregressive models. These paradigms, which progress

Table 7: **Preliminary Generalization of SSG to Other Architectures.** Generation quality (FID/IS) and inference efficiency (Steps/Time).

| Model | FID↓ | IS↑ | Steps | Time (s) |
|---|---|---|---|---|
| VQ-Diffusion | 9.39 | 158.3 | 100 | 7.3 |
| +SSG(**Adapted**) | **9.18** | **165.8** | 100 | 7.3 |

from noisy to clean representations or accumulate semantic information hierarchically, present natural anchors for guidance analogous to SSG. To empirically explore this concept, we performed a preliminary case study by applying an SSG-inspired formulation directly to the pre-sampling space of VQ-Diffusion (Gu et al., 2022). Evaluating metrics over 10,000 samples across 1,000 ImageNet classes at $256 \times 256$ resolution, our initial results in Tab. 7 demonstrate performance improvements. Specifically, SSG integration yielded a 0.21 reduction in FID and an 7.5 increase in IS, all while incurring negligible overhead to inference time. Despite the marginal improvement due to the conceptual and preliminary nature of this application, these findings strongly suggest that the theoretical establishment of SSG can indeed benefit broader paradigms exhibiting coarse-to-fine behavior, encouraging further research in this direction.

Table 8: **Latency Comparison of Models With and Without SSG.** ‡: Zero-padding replaces extrapolation from $L'_{\text{interp}}$.

| | Model | Without SSG | | | With SSG | | |
|---|---|---|---|---|---|---|---|
| | | mean | std | params | mean | std | params |
| 256x256 | VAR-d16 | 0.273 | 0.0303 | 310M | 0.279 | 0.0313 | 310M |
| | VAR-d20 | 0.320 | 0.0398 | 601M | 0.324 | 0.0288 | 601M |
| | VAR-d24 | 0.384 | 0.0288 | 1.0B | 0.390 | 0.0279 | 1.0B |
| | VAR-d30 | 0.530 | 0.0346 | 2.0B | 0.536 | 0.0372 | 2.0B |
| 512x512 | VAR-d36 | 1.28 | 0.0279 | 2.4B | 1.29 | 0.0326 | 2.4B |
| T2I | HART-d20 | 1.06 | 0.0280 | 732M | 1.07 | 0.0236 | 732M |
| | Infinity-2B | 1.83 | 0.0136 | 2.2B | 1.86 | 0.0125 | 2.2B |
| Ablations | VAR-d16(Nearest Neighbour) | – | – | – | 0.278 | 0.0297 | 310M |
| | VAR-d16(Linear) | – | – | – | 0.278 | 0.0278 | 310M |
| | VAR-d16(DSE‡) | – | – | – | 0.276 | 0.0317 | 310M |
| | VAR-d16(DSE with static $\beta_k$) | – | – | – | 0.279 | 0.0332 | 310M |
| Extension | VQ-Diffusion | 7.27 | 0.0893 | 594M | 7.27 | 0.0829 | 594M |

## J  LATENCY COMPARISON

We report wall-clock inference time (Tab. 8 and relative latency (Tab. 1, Tab. 2, Tab. 3, Tab. 4, Tab. 5, and Tab. 7). Due to VRAM limits on our available GPU (NVIDIA A6000), all reproduced measurements use batch size 1. Accordingly, table entries marked § (*reproduced*) are normalized to our locally measured VAR-d30 wall time at bs= 1, while entries without § use relative times taken from the literature, which are normalized to VAR-d30 as originally reported (typically at bs= 64) (Tab. 2 and Tab. 3). Thus, each relative time is computed against a VAR-d30 baseline measured under the same conditions as its source. The exact numbers can be found in Tab. 8

Especially, note that bs $= 1$ is applied only to VAR (across scales) for internal comparisons and for isolating the incremental cost of the SSG operation. This choice does not compromise validity: all entries remain comparable because each is normalized to a VAR-d30 baseline measured under matched conditions.

Results are averaged over 100 runs, reporting the sample mean (mean), standard deviation (std), and the model parameters (params) both before and after applying SSG.

## K    TEMPERATURE SCALING DETAILS

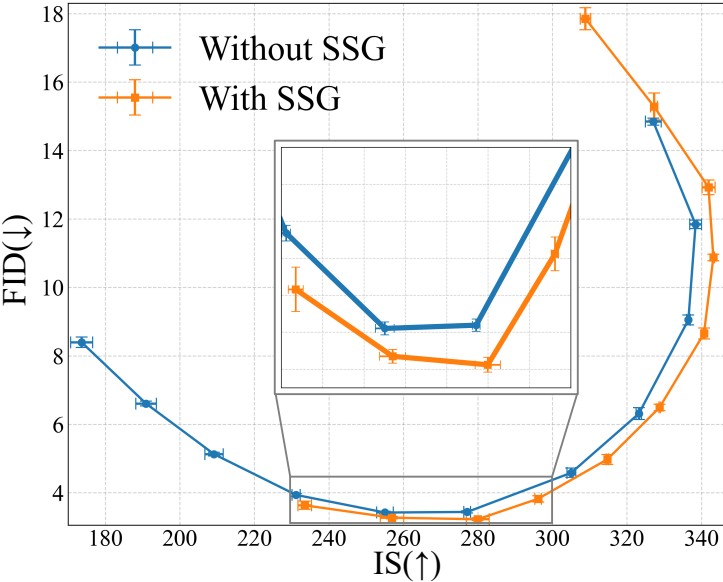

Figure 8: **Full-Scale FID vs. IS Trade-off.** This plot extends Fig. 4 (b) by showing the complete trade-off curves, averaged over 5 runs with error bars for both FID and IS. The curve with SSG consistently demonstrates a better quality-diversity profile, achieving both a lower minimum FID and higher maximum IS compared to the baseline across the full range of evaluated temperatures.

To ensure reproducibility for the results shown in Fig. 4 (b), we specify the temperature values used. For the baseline model (without SSG), we swept the temperature from 0.5 to 1.2. For our method (with SSG), we used a range of 0.7 to 1.5. Both evaluations were performed in increments of 0.1.

Figure 8 presents the full-scale FID vs. IS trade-off curve, which encompasses all data points used for Fig. 4 (b). This evaluation spans the temperature range from $0.5$ to $1.5$ in $0.1$ increments, yielding 11 data points in total. This plot explicitly includes the average of $N = 5$ independent runs across random seeds, with the uncertainty of both the FID and IS metrics indicated by error bars. As clearly observed in the full-scale result, the case with SSG (orange) demonstrates a superior trade-off profile than the baseline (blue) across the entire operational spectrum. The points achieved with SSG successfully form the Pareto frontier, attaining both the lowest FID and the highest IS on the curves. Crucially, the best FID recorded by our SSG is lower than the best baseline FID, with this substantial improvement falling outside the error bar range of the optimal baseline point. Furthermore, for any comparable data points, SSG consistently yields a better FID and IS, which robustly substantiates our initial claim that SSG provides a consistently better FID vs. IS trade-off.

## L    ADDITIONAL RELATED WORKS

**Diffusion models** are a central paradigm for visual generation (Ho et al., 2020; Nichol & Dhariwal, 2021). Early work such as latent diffusion (Rombach et al., 2022) employed U-Net backbones to iteratively denoise latent representations. While U-Nets provide strong multi-scale feature extraction, capturing long-range dependencies can be challenging, motivating transformer-based designs, such as DiT and U-ViT (Peebles & Xie, 2023; Bao et al., 2023). Transformers offer improved global interaction modeling and scale effectively, yielding fidelity gains with model size (Chen et al., 2024; Ma et al., 2024; Li et al., 2024a). Recent rectified-flow methods aim for faster, few-/single-step generation (Esser et al., 2024; Batifol et al., 2025), yet iterative denoising remains a major computational bottleneck in common pipelines, with substantial inference costs in memory and time (Peebles & Xie, 2023; Rombach et al., 2022; Yan et al., 2024; Hatamizadeh et al., 2024).

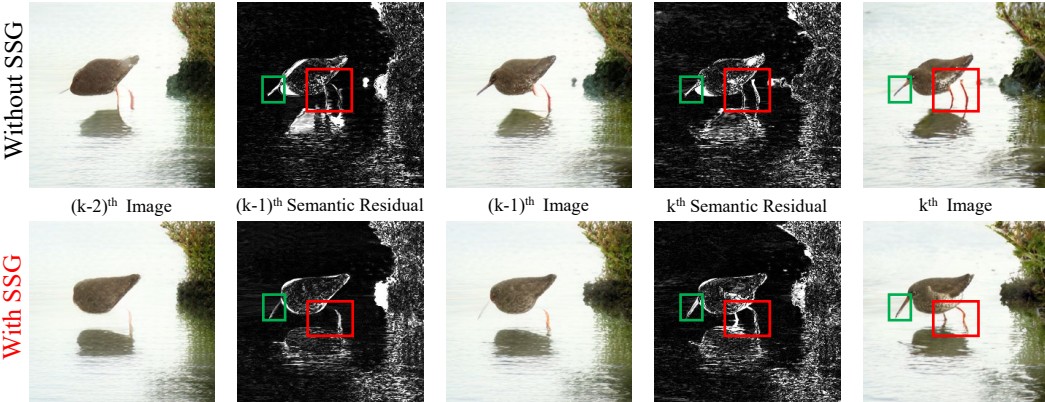

Figure 9: **Progressive Detail Enhancement with SSG.** Without SSG (top), the semantic residuals lack progressive detail, leading to artifacts like disconnected legs (red box). With SSG (bottom), the $k^{\text{th}}$ residual introduces finer, structurally coherent details, such as the clearer beak (green box) and properly connected legs (red box) not present at $(k-1)^{\text{th}}$, better realizing a coarse-to-fine nature.

## M  FURTHER QUALITATIVE COMPARISON ON FINE DETAIL GENERATION

This section provides a further qualitative examination of Fig. 9. Applying SSG not only adds fine detail but also improves overall visual coherence by placing those details consistently within the object structure, yielding more complete and perceptually stable entities.

We present additional qualitative evaluations of VAR models from d16 to d36 at 256×256 and 512×512 in the class-conditional setting. The results in Fig. 11 show that SSG consistently enhances fine detail and completes entities across VAR scales.

To further validate the improvements in generative quality in text-conditional generation using T2I models including HART (Tang et al., 2025) and Infinity (Han et al., 2025), we present additional qualitative evaluations. These results demonstrate that while SSG improves image fidelity, it also enhances the capability of these models to generate the precise details described in the input prompt. This is further illustrated in Fig. 12 and Fig. 13.

We also analyze failure cases where SSG yields limited improvements. Fig. 14 illustrates these limitations, categorized by (a) poor initial states and (b) challenging or ambiguous conditions. In column (a) (top), a VAR-d36 generation, SSG restores the main guitar structure but fails to render fine details including the strings. This is constrained by inherent model tokenization limits and a poor initial state with a severely distorted guitar that SSG cannot fully correct. For the HART model (middle), the prompt provides only vague screen-specific details. SSG offers minimal improvement, bounded by intrinsic model limitations in rendering this content, particularly when it is uncertain what to refine from the vague initial state. In the bottom example (HART), SSG successfully enforces the "a 7 year old brown skin girl" prompt detail and removes artifacts, yet fails to perfectly render the bird. This demonstrates that the corrective capability of SSG may be limited when starting from a severely misaligned initial state.

Column (b) in Fig. 14 highlights failures related to prompt comprehension or inherent ambiguity within a class. For class-conditional generation (VAR-d30, top), objects, such as the sea cucumber, that naturally fuse with the background are not distinctly generated. This occurs because SSG does not force such objects to be distinct, respecting their inherent nature to blend into the background. For the HART model (middle), given a highly ambiguous prompt like "Cosmic Death," SSG merely shifts the output from "Death" to "Cosmos" but cannot resolve the conceptual ambiguity, reflecting a model-level text understanding failure. Similarly, in the bottom HART example, the text encoder fails to parse specialized medical jargon, capturing only the word "goat". While SSG successfully removes most artifacts, it cannot compensate for the fundamental inability of the encoder to interpret the specialized prompt.

## N    HUMAN PERCEPTUAL EVALUATION

To validate the perceptual quality and semantic fidelity, we conducted a blind A/B choice human evaluation. This evaluation involved 28 participants, ranging from non-experts to experts in the visual generation field, who assessed 15 item pairs sampled across VAR-structured models, specifically VAR, HART, and Infinity. The results in Fig. 10 demonstrate a significant preference for the SSG-enhanced images. These images were favored in 71.0% of trials, compared to only 11.9% for the baseline. This robust subjective preference confirms that the superior spectral fidelity, coupled with stronger alignment to the given class or text

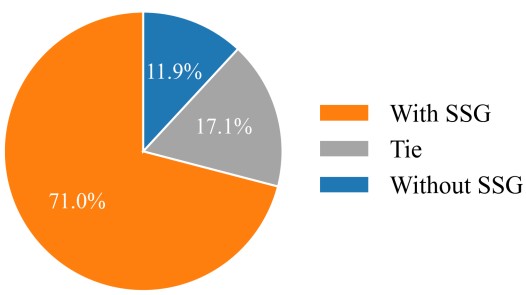

Figure 10: **Human Evaluation (A/B Test).** The SSG-enhanced method demonstrates superior perceived quality compared to the baseline

conditions, directly translates into a significant and robust improvement in perceived quality. The 17.1% tie rate indicates that the improvements provided by the SSG might be difficult to distinguish for non-expert evaluators in those instances, suggesting that SSG's enhancement often targets fine-grained details which, while objectively superior, require closer inspection to fully perceive.

## O    REPRODUCTION NOTES FOR REPORTED TABLES

We document the sources of all reported numbers. Unless otherwise noted, values in Tab. 1, Tab. 2, Tab. 3, and Tab. 4 are taken from the original papers. The mark § *reproduced* denotes results we computed due to issues with the released VAR pretrained weights (Tian et al., 2024); see Sec. 4.1 for details. For Tab. 4, all entries are our reproductions, due to problems detailed in Sec. 4.1.

## P    LIMITATIONS

SSG operates within the logit space during inference. Therefore, architectures that do not expose logits at the sampling stage, such as latent diffusion (Rombach et al., 2022) or flow matching (Lipman et al., 2023), which operate in continuous feature space by predicting noise or velocity vectors, respectively, require substantial modification to apply SSG, even though the underlying principle remains applicable.

## Q    THE USE OF LARGE LANGUAGE MODELS (LLMs)

We used LLMs solely for editorial assistance, to polish grammar mostly and converting paper-written mathematical expressions into LaTeX (including formatting proofs in the appendix). The model did not generate ideas, claims, or experimental content, and it was not used for data analysis or code design beyond minor formatting. All technical statements, equations, and results were authored and verified by the authors.

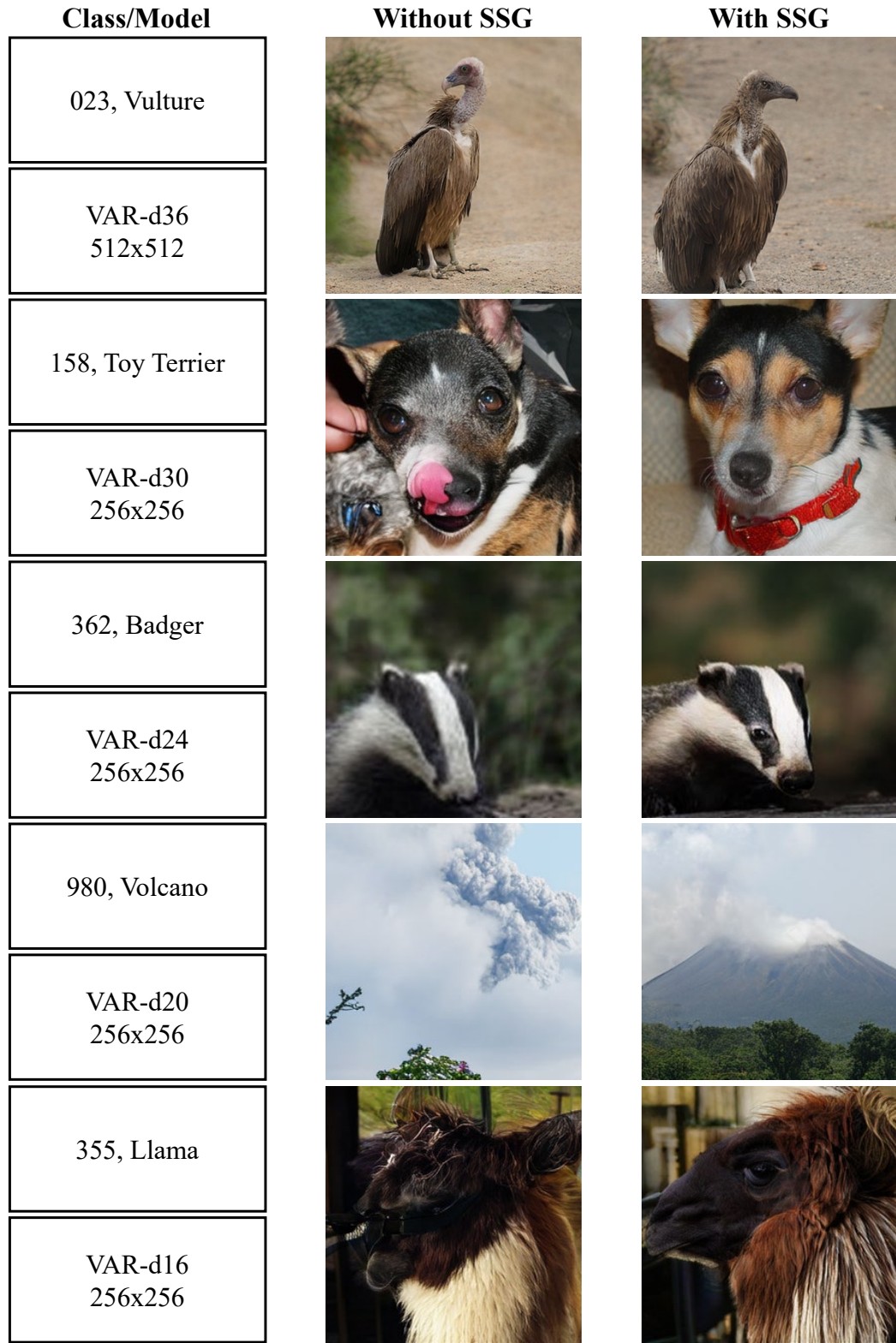

Figure 11: **Qualitative evaluation of VAR across scales.** Applying SSG enhances fine-detail generation consistently over multiple scales.

| Prompt | Without SSG | With SSG |
|---|---|---|

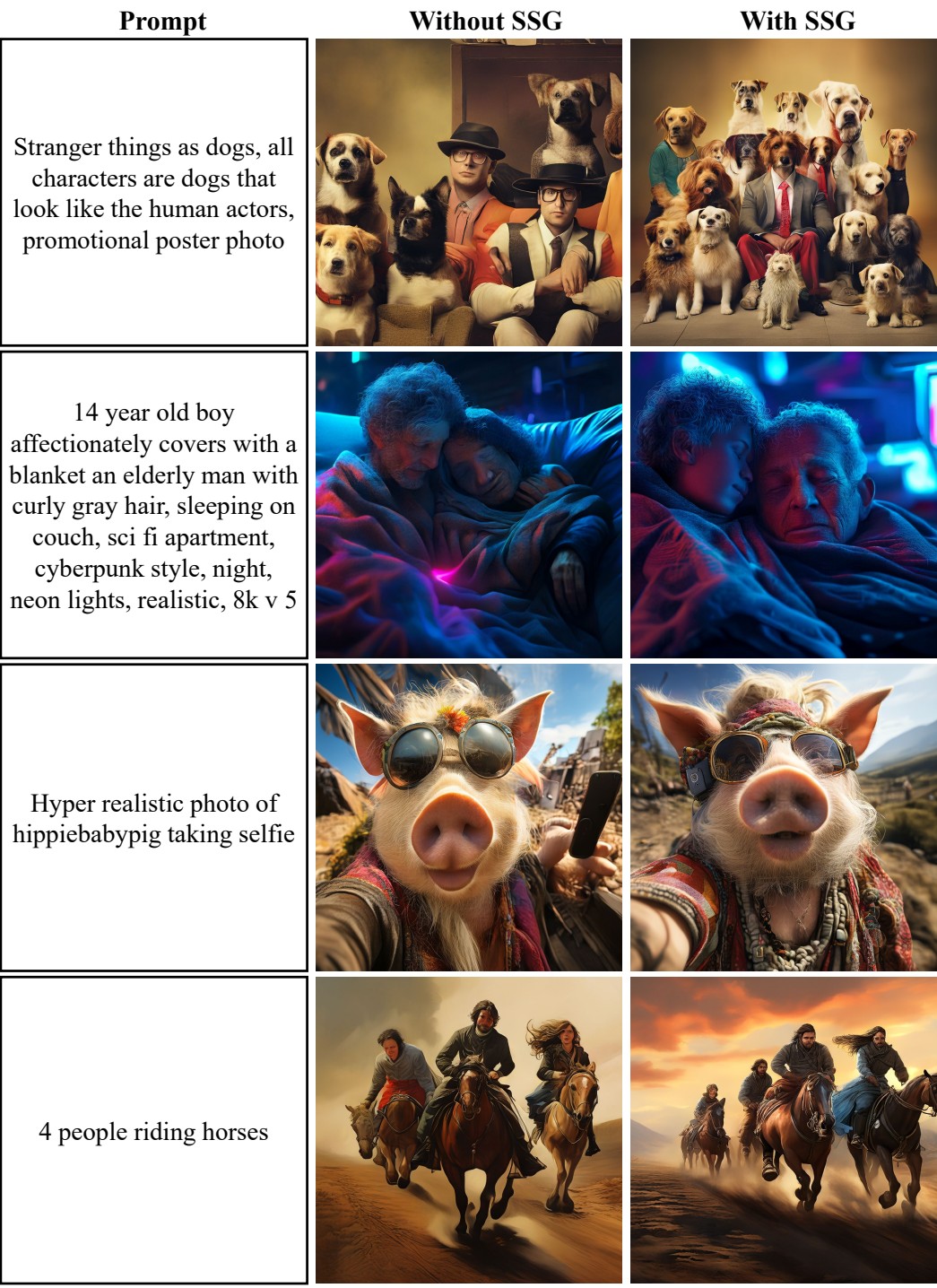

Figure 12: **Qualitative Evaluation using HART.** The use of SSG not only improves the quality of the generated images but also results in a stronger alignment with the input prompt.

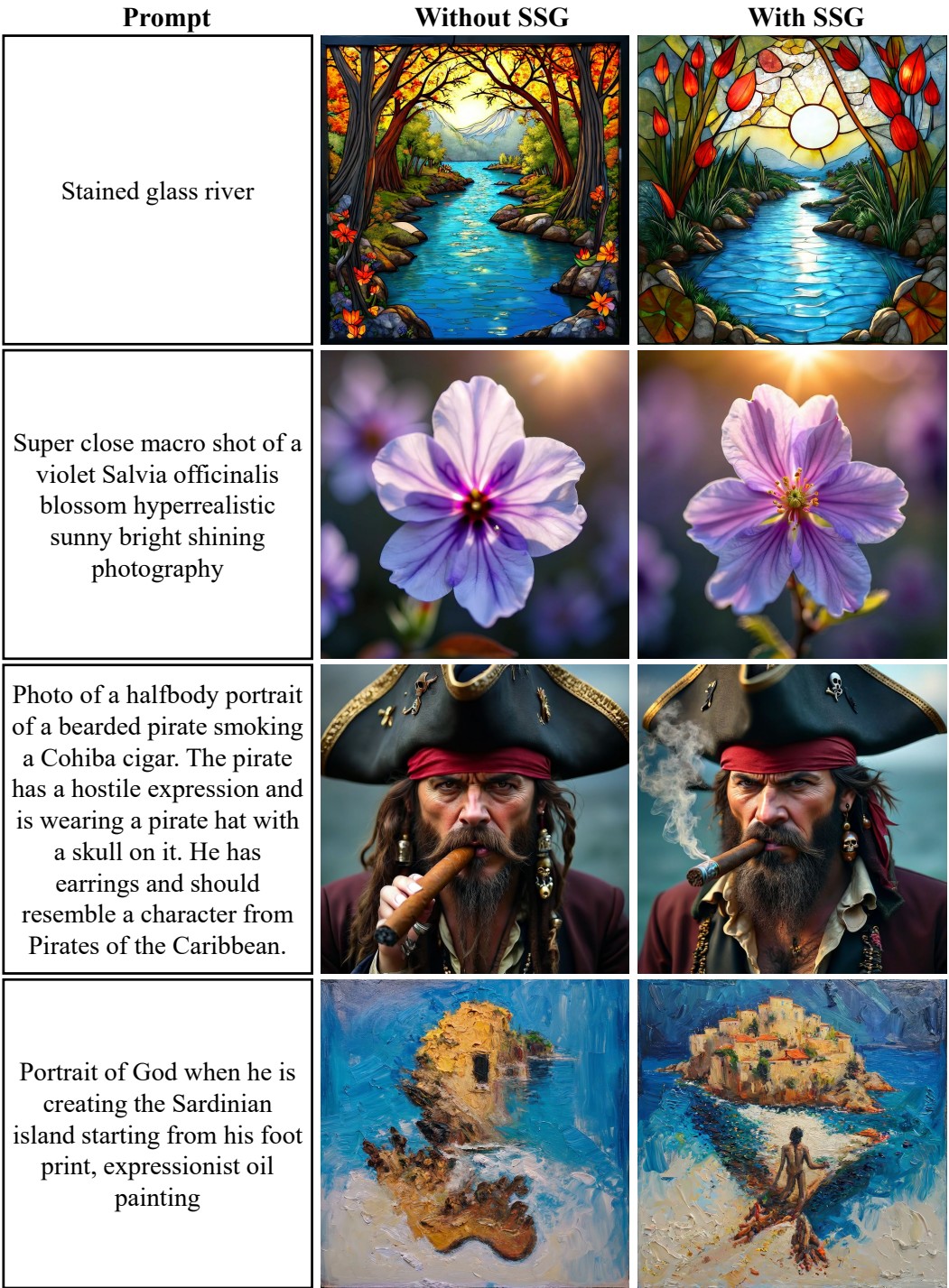

Figure 13: **Qualitative Evaluation using Infinity.** The use of SSG improves overall image quality. Most importantly, it captures the precise details depicted in the input prompt.

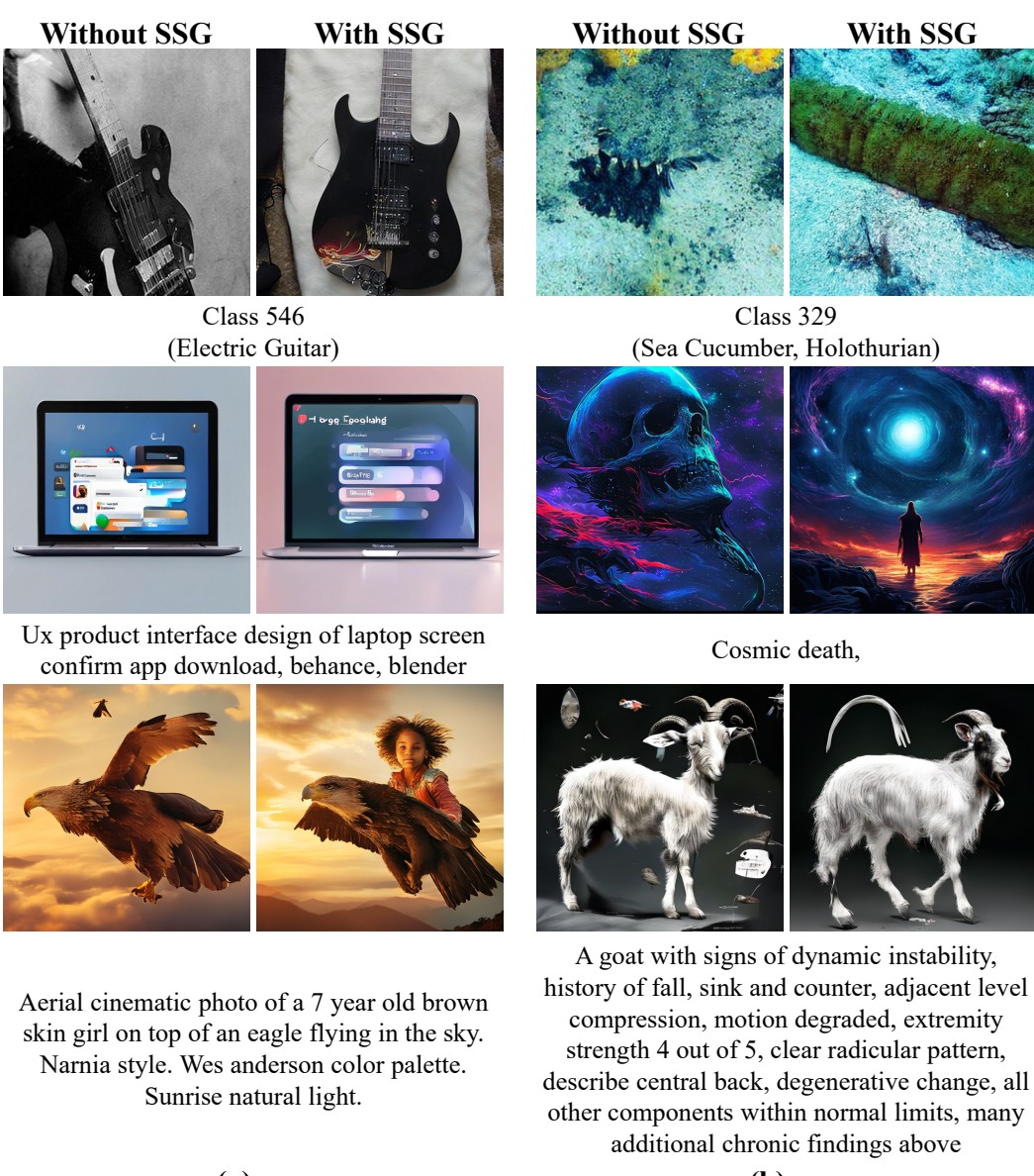

Figure 14: **Qualitative Evaluation on Failure Cases.** The corrective capability of SSG is bounded by initial states or task ambiguity. **(a)** Cases where SSG cannot fully recover from poor initial states stemming from tokenization issues or weak text-prompt alignment. **(b)** Limitations due to prompts being highly specialized or ambiguous, or when objects are inherently fused with the background.

