# OpenReview forum: "SSG: Scaled Spatial Guidance for Multi-Scale Visual Autoregressive Generation"
_ICLR.cc/2026/Conference — ICLR 2026 Poster_

### Official Review · Reviewer_oyJp · 2025-10-30

**Soundness:** 3
**Presentation:** 3
**Contribution:** 3
**Rating:** 6
**Confidence:** 4

**Summary:**

This paper proposes a training-free inference guidance method for Visual Autoregressive Generative Models (VAR), called Scaled Spatial Guidance (SSG). This method aims to address the problem of VAR deviating from the coarse-to-fine hierarchical generation pattern during inference. From an information theory perspective, the authors point out that each generation scale should only supplement high-frequency semantic residuals not covered by the previous scale. To this end, SSG guides the model's logits during inference by calculating residuals with the frequency domain prior constructed in the previous step and amplifying high-frequency components. Experiments show that SSG significantly improves image sharpness and structural consistency while maintaining the same inference speed.

**Strengths:**

1. The observations in this paper are highly relevant to the proposed method and make sense.

2. The proposed Discrete Spatial Enhancement effectively addresses the information loss caused by linear interpolation.

3. Comprehensive experimental data show that SSG can consistently improve the generation performance of various variable models without compromising the generation speed.

4. Ablation experiments validates the effectiveness of the proposed method.

**Weaknesses:**

1. Most of the experimental results in this paper come from class-conditional image generation. I believe that including more experimental results from text-to-image VAR models as the main experimental content would better demonstrate the practical value of the method. For example, adding more visual comparisons on infinity models to illustrate the visual improvement of the proposed method, or further expanding Table 4.

2. The proposed method seems to be applicable only to next-scale prediction.

**Questions:**

Please see the weakness

---

> ### Author Response · Authors · 2025-11-21
> **Response to Reviewer oyJp**
>
> We appreciate the reviewer’s time and the insightful questions raised. Below, we provide our responses to the specific points regarding qualitative results and method applicability.
> - **W1 - Insufficient qualitative results in Text-to-Image (T2I)**
>
>      We again appreciate the reviewer’s suggestion regarding the inclusion of additional text-to-image (T2I) experiments. In the main paper, we selected HART and Infinity as representative T2I VAR models to demonstrate that SSG remains effective not only in class-conditional settings but also in text-conditioned generation. Importantly, these models employ hybrid continuous–discrete tokens and bit-wise tokens, respectively, while still operating under a coarse-to-fine multiscale generation process. The objective of conducting experiments on these models was to show that SSG functions robustly across diverse tokenization strategies, rather than limiting the evaluation to the multi-scale vector-quantized tokenization used in VAR. We also included further qualitative examples in Appendix K (Fig. 12 and 13) to illustrate this behavior.
>
>     Nevertheless, we agree that expanding the T2I results could better highlight the practical value of our method. In the revised version of the paper, we have incorporated additional qualitative comparisons in Fig. 14 and provided a deeper analysis of SSG’s effects in text-conditioned generation in Appendix K, analyzing broader cases to better evaluate the impact of SSG. We would also like to note that Appendix E already reports extended quantitative evaluations on T2I Infinity, including ImageReward, HPSv2.1, and GenEval metrics, which we hope address some of the reviewer’s concerns regarding practical value.
>
> - **W2 - Applicability limited to next-scale prediction models**
>
>     Our proposed SSG method is intentionally designed around the coarse-to-fine generative structure of VAR models. The explicit multiscale hierarchy in VAR provides clear intermediate resolutions where logit manipulation can directly enhance per-step generation quality, naturally making SSG effective in next-scale prediction settings. Thus, the current algorithm is indeed specialized for VAR-style architectures.
>
>     However, the underlying information-theoretic principles that motivate SSG, including the frequency-domain interpretation of the pre-sampling logit space and the view that guidance can be understood as shaping the information flow before sampling, are not inherently limited to VAR. Generative models that exhibit a comparable coarse-to-fine progression, such as hierarchical autoregressive generators or diffusion models, may provide analogous stages at which SSG-inspired mechanisms could be applied. As an exploratory indication of this broader potential, we include preliminary experiments with VQ-Diffusion, showing that frequency-structured guidance can offer benefits even outside VAR:
>
>     | **Model** | **FID (↓)** | **IS (↑)** | **Steps** | **Time (s)** |
>     | --- | --- | --- | --- | --- |
>     | VQ-Diffusion (Baseline) | 9.386 | 158.29 | 100 | 7.3 |
>     | VQ-Diffusion (with SSG-inspired enhancement) | **9.182** | **165.83** | 100 | 7.3 |
>
>     Constructing fully developed extensions for these architectures would require substantial additional analysis and dedicated experimentation, which fall beyond the scope of the present work but represent promising directions for future research.

---

> > ### Comment · Reviewer_oyJp · 2025-11-24
> > **response to authors**
> >
> > After reading the comments from other reviewers and the authors' rebuttal, I believe this is a paper worthy of acceptance, and I have raised my score to 8.

---

> > > ### Author Response · Authors · 2025-11-25
> > >
> > > We sincerely appreciate your time in reviewing our rebuttal and the discussions with other reviewers, as well as your decision to raise the score. We are glad to hear that our additional qualitative results and the discussion on the broader applicability of our method alleviated your concerns. Thank you for your support and for recognizing the value of our work.

---

### Official Review · Reviewer_oX8V · 2025-10-31

**Soundness:** 3
**Presentation:** 3
**Contribution:** 3
**Rating:** 4
**Confidence:** 3

**Summary:**

This paper proposes a simple, training-free guidance trick for next-scale visual autoregressive (VAR) models called \emph{Scaled Spatial Guidance} (SSG). At sampling step $k$, the method constructs a coarse ``prior'' by upsampling the previous step's logits via a frequency-domain procedure (\emph{Discrete Spatial Enhancement}, DSE), and then applies a closed-form logit update
\[
\ell^{\text{SSG}}_{k} = \ell_{k} + \beta_{k}\bigl(\ell_{k} - \ell_{\text{prior}}\bigr),
\]
intended to emphasize the predicted high-frequency residual while keeping the coarse structure aligned with the autoregressive hierarchy. The approach requires no retraining, adds negligible overhead, and is reported to improve FID/IS across several VAR baselines and tokenizations on ImageNet (256/512) and MJHQ-30K, without increasing the number of sampling steps.


While practical and easy to bolt on, the novelty feels incremental relative to existing guidance/contrastive sharpening ideas (e.g., CFG-style logit shaping, spectral emphasis, residual boosting). The theoretical framing relies on strong heuristics---coarse-state Markov assumptions and a clean low/high-frequency separation in logit space---that may not hold for learned decoders; likewise, DSE implicitly assumes band-independence that is only approximate. The method appears sensitive to the quality of the transported prior and the schedule $\{\beta_k\}$ (and temperature), yet the paper offers limited analysis of failure modes (e.g., misaligned priors), robustness, or hyperparameter stability. Empirical scope is narrow (primarily ImageNet and a single T2I set), comparisons depend on in-house reruns under one protocol (raising fairness/reproducibility concerns), and there is no statistical testing, human preference evaluation, or detailed profiling of latency/memory across resolutions/hardware. The technique is appealing as an inference-time tweak, but stronger evidence (broader datasets, stress tests on compositional/OOD cases, ablations on DSE variants, instability/degeneration analyses, and transparent baseline parity) would be needed to elevate its contribution.

**Strengths:**

SSG is a one-line logit update with a clear algorithmic recipe (Alg. 1–2) and no retraining or architectural changes; it operates directly on residual logits and drops into a wide range of VAR-style decoders.

**Weaknesses:**

SSG feels close to existing guidance/contrastive logit-shaping (e.g., CFG-like sharpening, residual boosting), with limited conceptual leap.

The motivation for DSE/SSG leans on orthonormal transforms yielding “independent and non-interfering” bands, enabling “exact, lossless reconstruction” and a clean separation of low/high frequencies for prior construction. That’s a strong assumption in learned logit spaces and likely violated by aliasing, context coupling, and tokenization quirks; the paper doesn’t test how departures from this assumption affect outcomes.

Ablations show nearest/linear spatial priors worsen both FID and IS versus baseline (no SSG). This suggests SSG can hurt quality unless the frequency-domain prior is used, narrowing practical robustness and increasing configuration risk. Yet the paper doesn’t map when a user might inadvertently choose a weaker prior (e.g., for speed).

The method explicitly depends on the transported/upsampled prior. Misalignment (e.g., spatial drift, aliasing, or tokenization artifacts) can produce oversharpening, texture hallucinations, or suppression of semantically important low-frequency structure.

**Questions:**

In Fig. 4(b), did you sweep identical temperature grids for baseline and +SSG, and report matched points with CIs over multiple seeds? If not, can you replot with a shared grid and error bars to substantiate the “consistently better FID–IS trade-off” claim?

---

> ### Author Response · Authors · 2025-11-21
> **Response to Reviewer oX8V (1/4)**
>
> Thank you for your insightful and detailed review of our paper. We appreciate the time and effort you dedicated to assessing our work and providing valuable feedback, especially given the seriousness of the concerns raised regarding novelty, empirical rigor, and theoretical assumptions. We sincerely believe that the comprehensive revisions we have undertaken directly address these concerns and demonstrate the robustness and novelty of our contribution.
>
>
> Before proceeding to the specific point-by-point responses, we would like to offer initial clarifications and direct you to the new evidence in the revised manuscript that addresses the major concerns raised in your summary:
>
>    - **In-house Reruns (Fairness/Reproducibility):** The necessity for our in-house reruns was driven by the documented discrepancies between the original paper's reported values and those obtained using the publicly provided pre-trained weights. These discrepancies have been noted by the model authors in their public documentation and further discussed in publicly available issue threads. Our rationale for these reruns, along with the specific protocols used to ensure transparent and reproducible baseline comparisons, is fully detailed in Sec. 4.1 in the paper.
>
>    - **Empirical Scope and Generality:** We assure the reviewer of our methodological rigor. Our evaluation included three representative, publicly available models that span diverse tokenization methodologies (multi-scale VQ, discrete-continuous hybrid, and bit-wise), thereby robustly demonstrating generality across tokenization, which is a primary claim of the paper. This scope, anchored by ImageNet and a high-quality T2I dataset, is consistent with the empirical breadth used by many other representative visual synthesis papers to establish generality.
>
>    - **Human Preference Evaluation:** We have addressed the need for human intention alignment by including a human preference evaluation, which is now available in Appendix L of the revised paper.
>
>    - **Latency and Memory Profiling:** We direct attention to the existing detailed latency and memory analysis (presented in Tab. 8 and Appendix I), which provides profiling across resolutions for various VAR-structured models.
>
>    - **Analysis of Failure Modes/Robustness:** We have added an exploration of failure case analysis, including a discussion of the underlying conditions that appear to hinder performance, which is provided with Fig. 14 and Appendix K in the revised version.
>
> Throughout our detailed point-by-point response, we will address your concerns regarding the theoretical foundation and incremental novelty to clearly differentiate SSG from existing guidance mechanisms. We sincerely hope this evidence, coupled with our further analysis, will demonstrate the completeness and rigor of this paper.

---

> ### Author Response · Authors · 2025-11-21
> **Response to Reviewer oX8V (2/4)**
>
> - **W1 - Is SSG close to existing guidance/contrastive logit-shaping?**
>
>     We acknowledge the reviewer’s observation regarding the structural similarities between our mathematical formulation of SSG, which enhances the current prediction by leveraging the residual from the previous step, and existing guidance frameworks like **CFG** (Ho & Salimans, 2022), **Autoguidance** (Karras et al., 2024), and self-correction methods like **PAG** (Ahn et al., 2024), **SAG** (Hong et al., 2023). However, while the appearance may seem analogous to these, SSG presents a significant conceptual leap in mechanism, as it is specifically tailored for Visual Autoregressive Models (VAR) by adapting to their inherent nature.
>
>     The conceptual leaps are in the following list:
>
>      - **Self-Reference vs. Constructed Negative Branches**: Standard guidance methods (CFG) or self-correction techniques (Autoguidance, PAG, SAG) typically follow a contrastive philosophy that requires an explicitly generated unconditional or perturbed branch to serve as a negative reference. Unlike these methods, SSG eliminates the need for constructing a separate branch. Instead, it employs a mechanism of self-reference; it brings the previous step’s logit into calculation, which is already predicted in the VAR process and plays the role of an intrinsic reference signal. Thus, SSG achieves the sharpening effect of guidance without the computational overhead of dual inference passes. Also, unlike the existing guidance methods like autoguidance, PAG, SAG which operates on continuous noise or feature spaces in denoising process, SSG directly addresses the discrete logit probabilties which are not analgous and require conceptual leap for application.
>
>     - **Inter-scale vs. Intra-scale Guidance:** Methods including Autoguidance and PAG derive guidance signals by perturbing the current state to formulate a "bad" version of itself. This intra-scale perturbation can be limited because naive manipulation of the present step may introduce unreliable information or artifacts. In contrast, SSG introduces inter-scale hierarchical guidance. Instead of artificially perturbing the current scale to create a negative reference, SSG utilizes the previous scale’s logits as a natural structural anchor. By defining the residual between the current scale and the upsampled coarse scale, SSG sets a clear target for the new, high-frequency details emerging at scale k. This mechanism aligns naturally with the coarse-to-fine nature of VAR.
>
>     - **Distinction from Residual Boosting**: If the reference to residual boosting implies methods like **FreeU** (Si et al., 2024), we acknowledge that SSG shares the goal of reinforcing high-frequency details. However, our approach constitutes a distinct mechanism. While FreeU functions primarily as a frequency modulation filter that explicitly re-weights feature activations, SSG leverages the intrinsic hierarchical structure of autoregressive modeling for precision guidance. Instead of applying an arbitrary frequency gain, SSG utilizes the residual between scales to isolate the specific spectral innovation required at that resolution. This allows SSG to guide the next-token prediction probability directly in the logit space to match the true data distribution, rather than merely acting as a static sharpening filter.
>
>     Thus, although SSG leverages the established principles of existing guidance and contrastive logit-shaping, it constitutes an original adaptation to transfer such concepts to the hierarchical autoregressive domain of VAR, encouraging VAR to align better with its intrinsic coarse-to-fine nature. We hope the analysis and experimental results presented in this work can provide insight into the operation and future study of VAR-style generative models utilized across various fields, including MLLMs (Multi-modal Large Language Models), and offer a new perspective for enhancing performance in diverse visual generative models.

---

> > ### Author Response · Authors · 2025-11-21
> > **Response to Reviewer oX8V (3/4)**
> >
> > - **W2 - Addressing concerns regarding spectral independence and lossless reconstruction assumptions**
> >
> >     The reviewer's concern regarding the potential violation of the independence assumption in learned logit spaces due to aliasing or context coupling is certainly valid. However, although the perfect orthogonality found in analytical signal processing may not be guaranteed in this context, our empirical results demonstrate that SSG effectively achieves the desired spectral separation in practice.
> >
> >     As shown in Fig. 4(a) in the paper, we analyzed the spectral energy difference ($\Delta$ Log Magnitude) between the baseline and SSG across generation steps. The vertical red dashed line indicates the Nyquist frequency of the previous scale ($k$). These graphs substantiate our assumptions:
> >
> >      1. **Non-Interference in Base Frequencies ($f < f_{Nyquist}$):** To the left of the Nyquist boundary, the spectral difference remains close to zero. This indicates that SSG does not disturb or distort the low-frequency structure established by the previous scale, effectively confirming that there is no significant aliasing or interference propagating back into the base content.
> >
> >      2. **Targeted Innovation in High Frequencies ($f > f_{Nyquist}$):** To the right of the boundary, we observe a sharp positive increase in magnitude. This confirms that SSG selectively boosts the new high-frequency components required for the current resolution step.
> >
> >     Therefore, while the theoretical assumption of orthogonality in logit space is an approximation, Fig. 4(a) confirms that it is operationally valid: SSG successfully decouples the preservation of global structure from the generation of local details without destructive interference.
> >
> > - **W3 - Does the reliance on DSE limit practicality?**
> >
> >     The reviewer’s observation regarding naive spatial upsampling priors (Linear or Nearest Neighbor) is entirely correct: they perform worse than the baseline in both FID and IS. We would like to clarify that this result is, in fact, the central motivation for one of our primary contributions: **Discrete Spatial Enhancement (DSE).**
> >
> >     Our ablation study in Tab. 5 was designed to demonstrate exactly why these classical spatial methods are insufficient. As the reviewer correctly intuits, SSG functions by emphasizing new, high-frequency information relative to the previous step of the generation. Consequently, if SSG is provided with a **weak** prior containing the interpolation artifacts commonly found with naive linear interpolation, the model is misguided into amplifying those very artifacts, resulting in the poor performance evident in Tab. 5. This is precisely why our DSE prior is necessary. DSE utilizes a frequency-domain construction (such as DCT) that mathematically conducts an exact, lossless reconstruction via its transform and inverse transform. This process minimizes information loss when constructing the prior, and its practical robustness is validated by the superior results over naive linear interpolation in our ablation studies.
> >
> >     Regarding the **configuration risk**, we hope Tab. 8 alleviates this concern. The latency difference between using Linear interpolation (0.278s) and our DSE (0.276s) is negligible and well within the standard deviation of each measurement. Given that there is also no difference in memory burden, there is no practical incentive in terms of speed or memory for a user to choose naive, artifact-prone priors over our frequency-domain DSE, which produces the optimal result.

---

> ### Author Response · Authors · 2025-11-21
> **Response to Reviewer oX8V (4/4)**
>
> - **W4 - Robustness against Misalignment, Oversharpening, Texture Hallucinations, and Low-Frequency Suppression**
>
>     The concerns due to our reliance on prior is appreciated to give us opportunity how we addressed those problems that potentially raise the issues in generation, as they are what SSG was carefuly designed to navigate. We would like to go through the concerns you raised.
>
>     We appreciate the reviewer regarding the reliance on the prior. These potential pitfalls, including aliasing, oversharpening, and drift, were central considerations during the design of SSG. We address each specific concern below:
>
>      - **Aliasing and Low-Frequency Suppression:** It is true that misalignment or aliasing could theoretically damage semantically important low-frequency structures. However, our spectral analysis in Fig. 4(a) demonstrates that even if such aliasing occurs, it does not detrimentally impact the global structure in practice. The results show that SSG preserves the global structure established in the previous step without disturbance. Specifically, the spectral difference in the low-frequency range is negligible, empirically confirming that SSG does not suppress existing low-frequency content nor create destructive interference.
>
>      - **Oversharpening, Tokenization Artifact and Texture Hallucinations:** This is also a valid concern. However, while SSG emphasizes high-frequency detail, its sole purpose is not to recklessly increase spectral energy. To further substantiate our claim, we have added a new spectral analysis graph as Fig. 5 in the revised paper. Referring to this figure, if oversharpening or amplifying artifacts were taking place, the plot for SSG (orange line) should increase far beyond the black line (the reference spectral fidelity). Instead, the plot clearly illustrates that SSG consistently adds spectral energy in a manner that adheres better to the true distribution. It improves adherence to prevent over-amplification, thus functioning as a correction rather than a naive boost. Furthermore, the graph in Fig. 4(b) presents a better Pareto frontier for IS and FID, strongly substantiating that SSG does not create the non-semantic texture hallucinations that are detrimental to perceptual quality scores, including IS and FID.
>
>      - **Spatial Drift:** We clarify that spatial drift is not a significant risk because the prior is not an external, possibly misaligned signal; it is derived directly from the model's own previous generation step. This ensures inherent structural alignment between the current logit and the prior. Additionally, the improved sFID scores in Tab. 1 further validate that SSG generations adhere more closely to the true spatial distribution, a result that would be unattainable if significant spatial drift were present.
>
> - **Q1 - Full temperature range analysis of Fig. 4(b)**
>
>     In Fig. 4(b), we presented only a subset of the temperature sweep graphs, selecting data points that showed the most comparable FID vs. IS trade-offs between the baseline and the SSG-enhanced result. As requested by the reviewer, we have now provided the full-range grid and error bars in Fig. 10. This result was generated over five trials with random seeds across the full temperature spectrum (from 0.5 to 1.5 in 0.1 increments, encompassing 11 data points). As clearly observed in the full-scale result, the case with SSG (orange) demonstrates a superior trade-off profile to the baseline (blue) across the entire operational spectrum. The points achieved with SSG successfully form the Pareto frontier, attaining both the lowest FID and the highest IS on the curves. Crucially, the best FID recorded by SSG is lower than the baseline's best FID, with this substantial improvement falling outside the error bar range of the baseline's optimal point. This robustly substantiates our initial claim that SSG provides a consistently better FID vs. IS trade-off.

---

> > ### Comment · Reviewer_oX8V · 2025-11-27
> >
> > After reading the rebuttal and other reviews, I raise my score to 6.

---

### Official Review · Reviewer_YsKt · 2025-11-01

**Soundness:** 3
**Presentation:** 4
**Contribution:** 3
**Rating:** 6
**Confidence:** 4

**Summary:**

This paper introduces Scaled Spatial Guidance (SSG) — a training-free, inference-time enhancement for visual autoregressive (VAR) models. VAR models generate images via next-scale prediction in a coarse-to-fine hierarchy but often suffer from train–inference discrepancies, where fine scales redundantly reconstruct coarse details instead of adding new high-frequency content. SSG addresses this by guiding each generation step to focus on the semantic residual—the high-frequency information not captured by previous scales. This is achieved through Discrete Spatial Enhancement (DSE), a frequency-domain prior construction that preserves coarse structure while isolating finer details. Applied directly to logits during inference, SSG improves image fidelity and diversity across multiple VAR baselines (VAR-d16 to VAR-d36), achieving up to 11.5% FID improvement at 512×512 resolution with negligible latency.

**Strengths:**

1. SSG works purely at inference on logits, requiring no retraining, fine-tuning, or architectural changes.

2. The method is derived from an information bottleneck perspective, offering a principled justification for why emphasizing high-frequency residuals improves coarse-to-fine generation.

3. The frequency-domain DSE module is simple and elegant.

4. Performance is good. SSG consistently improves VAR across various model sizes and input resolution, with only negelectable cost.

**Weaknesses:**

1. The proposed method is closely tied to the coarse-to-fine next-scale structure of VAR models. While this focus is well-motivated, it somewhat limits the generality of the contribution. It remains unclear whether SSG can extend to broader autoregressive or diffusion-based generation frameworks. Discussing how the information-theoretic insights or the frequency-domain prior could inspire guidance mechanisms beyond VAR would strengthen the paper’s broader impact.

2. Most examples highlight success cases where SSG clearly improves detail. Including a few failure or neutral cases could offer a more balanced understanding of when the guidance might be less effective.

**Questions:**

See Weaknesses.

---

> ### Author Response · Authors · 2025-11-21
> **Response to Reviewer YsKt**
>
> We sincerely thank the reviewer for the valuable feedback. We particularly appreciate the suggestions to discuss the applicability of our method beyond VAR and to explicitly analyze failure cases, as these additions have significantly improved the balance and depth of our manuscript.
>
> - **W1 - Applicability of SSG beyond VAR**
>
>     Our proposed SSG is intentionally designed to adhere to the coarse-to-fine generative nature of VAR-structured models. However, the underlying information-theoretic perspective introduced in Section 2.2, which interprets the pre-sampling logit space with frequency-domain structure to formulate guidance in a coarse-to-fine generative manner, is not inherently restricted to VAR. In principle, SSG-inspired mechanisms could extend to other generative families.
>
>
>     Diffusion models, for example, follow a progression from noisy to clean representations that can be viewed as a form of coarse-to-fine refinement, and hierarchical autoregressive models accumulate increasingly rich semantic information across their hidden states. These intermediate stages may provide potential anchors for guidance strategies analogous to SSG. However, implementing a fully operational analogue of SSG within these architectures would require substantial analysis of their pre-sampling spaces and extensive empirical verification, both of which lie outside the scope of the present work. We regard constructing concrete, architecture-appropriate formulations for these broader generative paradigms as a promising direction for future exploration.
>
>     Nonetheless, we would like to share our preliminary experiment to examine the potential viability of implementing our SSG method, coupled with our frequency-domain prior insight, in architectures other than VAR. We chose VQ-Diffusion as it is structurally differentiated from VAR models, allowing us to test the broader applicability of our conceptual contributions.
>
>     | **Model** | **FID (↓)** | **IS (↑)** | **Steps** | **Time (s)** |
>     | --- | --- | --- | --- | --- |
>     | VQ-Diffusion (Baseline) | 9.386 | 158.29 | 100 | 7.3 |
>     | VQ-Diffusion (with SSG-inspired enhancement) | **9.182** | **165.83** | 100 | 7.3 |
>
>     These experimental results validate our analogy that interpreting the denoising process as a coarse-to-fine refinement within an information-theoretic framing has potential viability. This suggests a possibility for expanding guidance applications beyond VAR-structured architectures and encourages further investigation into their applicability across different generative frameworks.
>
> - **W2 - Necessity for Failure/Neutral cases**
>
>     Before sharing some of our failure and neutral cases, we would like to express our appreciation for the reviewer’s suggestion to include such non-optimal cases. While the main text focuses on representative scenarios where SSG yields clear improvements, our experiments also reveal situations in which the guidance provides limited gains or introduces undesirable results. To provide a more balanced perspective on the strengths and limitations of SSG, we added several of these neutral and failure cases in the revised version of the paper, specifically in Fig. 14, along with brief analyses of the underlying conditions that appear to reduce the effectiveness of the guidance in Appendix K. In summary, our analysis highlighted specific extreme cases where performance degradation occurred, often attributable to inherent limitations of the tokenizer or ambiguity within the conditioning information for generation, factors that appear to hinder the performance of SSG.

---

### Official Review · Reviewer_snkW · 2025-11-05

**Soundness:** 3
**Presentation:** 3
**Contribution:** 3
**Rating:** 6
**Confidence:** 5

**Summary:**

This paper proposes a training-free guidance mechanism for Visual Autoregressive (VAR) models by leveraging the inherent scale-structured generation order. Inspired by information bottleneck principles, the authors introduce Scaled Spatial Guidance (SSG), which adjusts token generation dynamics across scales to correct distortions and enforce structure. Experiments across multiple architectures and benchmarks demonstrate consistent improvements in synthesis quality, suggesting that scale-aware inference adjustments can serve as an effective plug-and-play enhancement for VAR models.

**Strengths:**

- The paper identifies that the generation order of VAR models implicitly forms a scale-guidance structure and introduces a training-free guidance method that exploits this property.
- The proposed approach is grounded in the information bottleneck perspective, and the authors empirically support that the method can correct distorted signals.
- Extensive experiments across various models and datasets demonstrate consistent improvements, highlighting the robustness and generality of the method.

**Weaknesses:**

- While the method appears generic, the presentation is heavily specialized for VAR, making the broader applicability unclear.
- The approach, although training-free, feels ad-hoc and may not fundamentally address scalability and representation issues in visual tokenization; it would strengthen the contribution to explore how SSG could be incorporated into tokenization or model design directly.
- The abstract emphasizes improving high-frequency details, but the validation for this claim is limited and not rigorously demonstrated.

**Questions:**

**On generalization of the method**
Could the proposed method extend beyond VAR to other hierarchical generative frameworks such as diffusion guidance or hierarchical autoregressive models in below? If not directly, what modifications might be required?
- Parallel Multiscale Autoregressive Density Estimation, ICML 18
- Generating Diverse High-Fidelity Images with VQ-VAE-2, NeurIPS 19
- Locally Hierarchical Auto-Regressive Modeling for Image Generation, NeurIPS 22

**On frequency-domain evidence**
The paper claims enhanced high-frequency fidelity (e.g., Fig. 7), but the evidence is mostly visual. It would be beneficial to provide spectral analysis:
- Compare the average spectral amplitude of the dataset vs. VAR w/o SSG vs. VAR+SSG.
- Test Gaussian blur sensitivity: if SSG enhances high-frequency components, performance should degrade more under blurring.
- The paper On the Frequency Bias of Generative Models (NeurIPS'21) may serve as a useful analytical framework.
- Adding spectrum visualizations alongside Fig. 8/9/10 could provide direct support.

**On Appendix L temperature settings**
Appendix L reports different temperature ranges for baseline w/ and w/o SSG (0.5–1.2 vs. 0.7–1.5). Would it be fairer to evaluate both methods over the same temperature range (e.g., 0.5–1.5) and report best results or display curves for each? Since temperature sweeps were performed, sharing plots would clarify any performance variance across sampling strengths.

---

> ### Author Response · Authors · 2025-11-21
> **Response to Reviewer snkW (1/3)**
>
> We appreciate the time the reviewer have taken to thoroughly evaluate our paper and for raising the concerns and questions. We will take this opportunity to address your concerns and questions, beginning with the weaknesses you pointed out.
>
> - **W1 - Unclear broader applicability beyond VAR**
>
>     The proposed method, SSG, is designed for coarse-to-fine generative architectures. Our paper primarily focused on elevating the quality of models leveraging the efficient yet powerful generative potential of VAR; thus, the specified algorithm is directly applicable to VAR-structured models, including VAR, HART, and Infinity, as mentioned in the paper.
>
>     However, the core principle of manipulating the pre-sampling space with logits is more general. We have re-evaluated the theoretical groundwork established in Section 2.2 to consider its application to other generative paradigms. The information-theoretic perspective of improving per-step quality insight can benefit the iterative denoising process in diffusion models. This is because the diffusion process, moving from a highly noised state to a clean state, can be interpreted, to an extent, as a coarse-to-fine generative framework analogous to VAR.
>
>    While a direct application is not feasible, as diffusion's generative mechanism differs significantly from the expanding token maps of VAR, as an exploratory effort, we have tailored the per-step improvement insight for an iterative diffusion architecture. Specifically, we applied this approach to VQ-Diffusion, and the detailed quantitative results will be presented in our response to Q1.
>
> - **W2 - Concerns regarding scalability, tokenization representation, and the ad-hoc nature of the approach**
>
>     Our method, SSG, is a guidance mechanism applied in the pre-sampling logit space, designed to improve the sampling process and generate perceptually improved images agnostic to the specific tokenization process or model design. Our experimental results specifically show that SSG is effective across diverse tokenization methods, including multi-scale Vector Quantized, discrete-continuous hybrid, and bit-wise tokenization. Although our research focused on improving performance on already available pre-trained models and tokenizers, limiting the scope of tokenizer scale testing, the scalability analysis for the model itself is thoroughly presented in Tab. 1 of the paper.
>
>    We respectfully posit that SSG is not ad-hoc. Attempts to improve performance on pre-trained models by altering or enhancing sampling methods with guidance, without modifying the architecture, represent an active research area in the generative model literature, exemplified by diffusion samplers like DDIM and guidance methods like CFG++. Our paper not only presents performance improvements but also establishes a rigorous theoretical analysis of the method grounded in the well-established Information Bottleneck (IB) principle. We believe this theoretical foundation and the potential it presents for research in this area clearly differentiate SSG from a purely ad-hoc solution.
>
>     We sincerely appreciate the reviewer's suggestion to explore how our information-theoretic perspective might directly address scalability and representation issues in visual tokenization and model design. Incorporating this perspective directly into the tokenization or model design phases certainly represents an intriguing and important research direction. However, exploring this integration goes beyond the scope of the present work, which focuses on a training-free guidance mechanism for pre-trained models. We thus designate this comprehensive study as a promising direction for future work.
>
>
> - **W3 - improving high frequency details in the abstract**
>
>     We thank you for raising this concern, which allows us to elaborate more rigorously on the pixel level. Although we provided qualitative demonstrations (Fig. 1, 11, 12, and 13) and spectral analysis at the latent space level (Fig. 4(a)) in the revised paper, we acknowledge that we lacked the pixel-level spectral analysis needed to fully support our claim. We wish to clarify that our claim of "improving high-frequency detail" is not simply about increasing spectral energy, an objective also achievable by adding unnecessary artifacts or noise, potentially compromising perceptual fidelity. Instead, our objective is to inject high-frequency details that improve the expressiveness of the generated image while remaining close to the target distribution. This ensures the gained details are properly attributed to the subject of generation, not unintended artifacts. We provide this additional pixel-level spectral analysis in our response to Q2.

---

> ### Author Response · Authors · 2025-11-21
> **Response to Reviewer snkW (2/3)**
>
> - **Q1 - On generalization of the method**
>
>     As a pioneering effort to extend our theoretical foundation to hierarchical generative frameworks, specifically to diffusion models, we adapted an SSG-inspired mechanism to VQ-Diffusion. This choice was motivated by the fact that the iterative denoising process can, to an extent, be interpreted as a coarse-to-fine framework analogous to VAR. We present the quantitative results below. These metrics were calculated using 10,000 samples generated for 1,000 Imagenet classes at a resolution of $256 \times 256$:
>
>     | Model | FID $\downarrow$ | IS $\uparrow$ | **Steps** | **Time (s)** |
>     | --- | --- | --- | --- | --- |
>     | VQ-Diffusion (Baseline) | 9.386 | 158.29 | 100 | 7.3 |
>     | VQ-Diffusion (with SSG-inspired enhancement) | **9.182** | **165.83** | 100 | 7.3 |
>
>     These results demonstrate the potential benefit that the information-theoretic guidance mechanism can bring to diffusion architectures, yielding improvements in both FID and IS metrics. However, we emphasize that diffusion models are inherently different from VAR-structured models. For instance, VAR focuses on causal generation through expanding token maps, while diffusion relies on an iterative denoising process governed by complex noise schedules. Rigorously establishing the optimal guidance for diffusion models, or for other hierarchical generative frameworks, necessitates further exhaustive examination of how noise schedules and related attributes impact the pre-sampling space.

---

> ### Author Response · Authors · 2025-11-21
> **Response to Reviewer snkW (3/3)**
>
> - **Q2 - On frequency-domain evidence**
>
>     To ensure a statistically robust evaluation, we analyzed the spectral energy profiles of the entire dataset (10,000 images from the ImageNet validation set used to calculate the metrics in Tab. 1 of the paper) compared to generated samples from VAR-d16 with and without SSG (50,000 images each). Utilizing the full set of generated images eliminates selection bias and provides a comprehensive view. The resulting graph, added to the revised paper as Fig. 5, demonstrates the effectiveness of SSG in enhancing high-frequency details while maintaining fidelity to the ground-truth distribution.
>
>     We specifically focus on the frequency range of $10^1$ to $10^2$, which corresponds to meaningful high-frequency details (textures and edges) rather than basic structure or extreme noise. In the range below $5.5 \times 10^1$, the SSG curve (orange) consistently exhibits higher spectral energy than the Baseline curve (blue), effectively enhancing meaningful details. Crucially, a distinction emerges beyond $5.5 \times 10^1$. Here, the Baseline curve (blue) significantly diverges from the ground-truth (GT) reference (black line), suggesting the possible amplification of high-frequency artifacts or noise. In contrast, the SSG curve maintains a tight alignment with the black curve. This demonstrates that SSG does not blindly amplify high-frequency components; instead, it regulates the generation process to maintain fidelity to the GT distribution.
>
>     This global statistical analysis serves as a more reliable alternative to the sample-wise spectral analysis for Fig. 8, 9, and 10 (Fig. 11, 12, and 13 in the revised version), which we omitted to prevent misinterpretation. As noted in *On the Frequency Bias of Generative Models (NeurIPS'21)*, spectral analysis is valuable but requires ground-truth (GT) images for a fair comparison at the sample level. Ideally, higher spectral energy should indicate rich details; however, in the absence of a paired GT, it can be indistinguishable from high-frequency noise. Since our samples are generated from class labels or text prompts (lacking pixel-wise GT), a direct sample-by-sample spectral comparison could misleadingly favor noisy outputs, whereas our global analysis confirms the distributional alignment.
>
>
>     We also conducted the Gaussian blur sensitivity test as requested by the reviewer. Theoretically, Gaussian blur acts as a low-pass filter; thus, if SSG truly enhances meaningful fine details rather than noise, the resulting images should exhibit greater sensitivity to blurring, leading to a steeper degradation in quality metrics. We applied a Gaussian blur of the same strength to both with and without SSG samples and measured the drop in FID and IS. As shown in the table below, the IS is more rapidly degraded for the generated samples with SSG. This finding indicates that SSG enhances meaningful high-frequency details rather than arbitrary noise. However, we acknowledge that the validity of this evaluation is inherently limited, as standard metrics like FID and IS are trained on high-fidelity, non-blurred image distributions. Therefore, understanding the high-frequency fidelity of SSG is more reliably achieved through the global statistical analysis presented above.
>
>     | **Method** | **FID (Original) ↓** | **FID (Blurred) ↓** | **Δ FID (Degradation)** | **IS (Original) ↑** | **IS (Blurred) ↑** | **Δ IS (Degradation)** |
>     | --- | --- | --- | --- | --- | --- | --- |
>     | Baseline (Without SSG) | 3.42 | 16.96 | +13.54 | 275.6 | 148.5 | -127.1 |
>     | Ours (With SSG) | 3.27 | 16.83 | +13.58 | 285.3 | 154.4 | -130.9 |
>
>
> - **Q3 - On Appendix L temperature settings**
>
>     We initially presented only a part of the temperature sweep graphs because those data points represented the most comparable FID vs. IS trade-offs between the baseline and the SSG-enhanced result. Addressing the reviewer's concern, we now present the full graph, covering the temperature range of 0.5 to 1.5 (with 0.1 increments, encompassing 11 data points), to eliminate potential bias. We also conducted five trials with random seeds and added error bars to enhance the statistical reliability of our results. The resultant graph is included in the revised paper as Fig. 10, with detailed analysis provided in Appendix M (formerly Appendix L). This full-scale graph consistently demonstrates a superior FID vs. IS trade-off for SSG, showing that the substantial improvement achieved by SSG falls outside of the error bar range of the baseline's optimal point, thereby confirming the superiority of our method.

---

### Author Response · Authors · 2025-11-21
**Revision Summary**

We first would like to thank all reviewers for taking their valuable time to thoroughly evaluate our paper and for giving us insightful and constructive feedback.

With the feedback on broader applicability, we were able to experiment with the applicability of SSG to the broader coarse-to-fine generative paradigm beyond VAR. Also, regarding the request for spectral analysis on high-frequency fidelity, we were able to further explore pixel-level analysis to clarify the precise meaning of improving high-frequency detail in the paper, demonstrating how SSG improves the fidelity of generated images. Regarding concerns about novelty, we took this chance to elaborate on how our method differentiates from other existing guidance methods and to further substantiate the information-theoretic foundation of our SSG formulation. We also substantiated some of the assumptions made in formulating SSG and DSE with exhaustive experimental results, including failure cases to analyze what potentially limits the performance of SSG. Finally, to address concerns regarding statistical testing and human evaluation, we have added them to further highlight the robustness of our experimental results.

By addressing these points, we added claims along with figures and tables in our revision to further improve the validity and reliability of our paper, as listed below:

- **Spectral Amplitude Analysis (Sec. 4.5, Fig. 5)**

    We added a spectral amplitude analysis that elaborates on the precise meaning of improving high-frequency detail and demonstrates how SSG better adheres to the target distribution.

- **Broader Applicability of SSG (Sec. 4.7, Tab. 6; Updated Tab. 8)**

    With a preliminary generalization of SSG to VQ-Diffusion, we show that SSG can extend beyond VAR-structured models. We also updated the latency comparison, confirming that the adapted SSG also introduces only negligible computational overhead.

- **Broader Analysis of SSG’s Impact (Appendix K, Fig. 14)**

    We added qualitative failure cases and a comprehensive analysis of what may potentially hinder the performance of SSG.

- **Human Perceptual Analysis (Appendix L, Fig. 9)**

    Through a blind A/B human evaluation, we exhibit that SSG synthesizes not only perceptually appealing images but also images that remain well-aligned with the input condition (class or text prompt).

- **Full-scale FID–IS Trade-off with Statistical Testing (Appendix M, Fig. 10)**

    We added a full-scale FID vs. IS trade-off plot with error bars and detailed the statistical protocol, ensuring fair, rigorous comparison and reducing potential bias.

We have marked all changes in blue in the revision to help readers easily locate the updates. We hope our revision alleviates the reviewers’ concerns and further strengthens the practicality and rigorousness of our method, SSG.

---

### Author Response · Authors · 2025-12-01
**Summary of Rebuttal Consensus and Review History**

Dear Area Chair,

Due to the unprecedented security incident and the subsequent reset of the review process, we are deeply concerned that the constructive consensus achieved during the discussion phase may be obscured. To ensure that the valuable and logical discourse between the reviewers and authors is preserved, we respectfully summarize the key conclusions reached, which remain visible in the text trails:

**Unanimous Recognition of Strengths**

Before the discussion phase began, all reviewers independently identified the core strengths of our work, forming a positive consensus on the proposed method:

   - **Methodological Soundness**: **Reviewers snkW** and **oyJp** praised the information-theoretic perspective and principled justification for realigning the coarse-to-fine nature of generation by correcting distorted signals within the VAR structure. Additionally, **Reviewer oyJp** highlighted the effectiveness of the Discrete Spatial Enhancement (DSE) module, which **Reviewer YsKt** described as "simple and elegant."

   - **Practical Efficiency:** **Reviewer oX8V** highlighted the one-line logit update and clear algorithmic recipe that integrates into VAR architecture without retraining or architectural modifications. **Reviewer YsKt** noted the method achieves significant FID improvements with negligible latency.

   - **Robustness:** All reviewers acknowledged the consistent improvements across various model scales and different VAR-structured architectures demonstrated in our extensive experiments.

**Resolution of Concerns and Final Consensus**

  - **Reviewer oyJp (Score: 6 $\rightarrow$ 8):** Explicitly stated, ***"I believe this is a paper worthy of acceptance, and I have raised my score to 8,"*** after reviewing our rebuttal. This consensus was reached as we addressed concerns regarding the broader applicability of our information-theoretic methodology in coarse-to-fine generative models. We empirically demonstrated that this potential is not constrained to VAR-structured models and can be further applicable to other hierarchical generative frameworks, providing quantitative validation via exploratory experiments on diffusion models. Additionally, we provided further qualitative results on T2I models to analyze the impact of our method, which we believe was a key factor in securing the reviewer's support.

  - **Reviewer oX8V (Score: 4 $\rightarrow$ 6):** Explicitly stated, ***"After reading the rebuttal and other reviews, I raise my score to 6."*** This update acknowledged our additional experiments, including the Human Evaluation and full-range temperature sweeping with statistical testing. It also confirms that we successfully resolved questions regarding fairness, reproducibility, and the novelty/robustness of our methodology: Scaled Spatial Guidance (SSG) and Discrete Spatial Enhancement (DSE).

   - **Reviewers snkW and YsKt (Score: 6):** These reviewers had already assigned positive scores falling within the acceptance range. Although the discussion phase was abruptly ceased, we rigorously addressed their remaining questions, which overlapped significantly with the points resolved for the reviewers above.

      - **Broader Applicability:** Both requested evidence of applicability beyond VAR-structured models. We addressed this by empirically demonstrating the promising potential of our method for broader hierarchical generative models, such as diffusion and other autoregressive architectures, using the same diffusion model experiments that successfully convinced Reviewer oyJp.

       - **Specific Technical Requests:** For **Reviewer snkW**, we provided pixel-level spectral analysis to demonstrate that SSG adds meaningful high-frequency spectral energy, alongside the rigorous statistical temperature sweeps that also convinced Reviewer oX8V. For **Reviewer YsKt**, we provided an exhaustive analysis of failure cases to illustrate SSG's behavior under non-optimal architectural conditions to provide a more balanced understanding of the scope of SSG.

We firmly believe that the valuable time and effort dedicated by the authors, reviewers, and organizers should not be obscured by these circumstances. The text trails confirm that **all four reviewers have either explicitly raised their scores or maintained positive scores within the acceptance range**, supported by our comprehensive reasoning in the responses. We hope this summary assists you in navigating the review history efficiently and alleviates any potential questions you may have as the Area Chair. However, should there be any further points requiring clarification, please let us know, as we stand ready to address any remaining concerns immediately. We sincerely hope that, despite these disruptions, the evaluation process remains valid and equitable for all submissions.

Sincerely,

Authors

---

### Meta-Review · Area_Chair_Yj4A · 2025-12-19

**Summary:**

The paper introduces a training-free, information bottleneck-inspired method to steer VAR-based image generation towards disentangling the frequency information generated at each scale.

Reviewers praised the theoretical grounding, simplicity and effectiveness of the method. All reviewers were concerned about the limited applicability that requires VAR-based models. Some concerns were about lack of empirical validation of some motivations.

Most reviews were positive with only oX8V recommending rejection, raising novelty concerns due to similarity to CFG and other methods. The rebuttal provided evidence to address most reviewers concerns, including oX8V's, so I recommend acceptance.

**Reviewer Concerns:**

The rebuttal provided evidence that the method is also applicable to VQ-Diffusion, but in a specific setting that is far from SoTA performance, so it partially addressed reviewers concerns, although I don't believe this should prevent acceptance.

The rebuttal also added more analysis of the generated spectra, failure cases, and more text-to-image experiments, mitigating many reviewer concerns.

oX8V's concern about novelty was addressed by noting that CFG and other methods need modifying both the training procedure (by adding the "unconditional" class) and two passes per step during inference, while the proposed method requires no training changes and only need to look at previously generated logits for guidance, which is a significant conceptual difference. Their concern about DSE was also addressed thoroughly.

**Reviewer Scores:**

oX8V's concerns were addressed satisfactorily so I think they would increase their score. The other reviewers were already positive and also had most concerns addressed so they would maintain or increase their score.

---

### Decision · Program_Chairs · 2026-01-26

Accept (Poster)